# From net-zero to zero-fossil in transforming the EU energy system

Felix Schreyer [1,2] ✉, Falko Ueckerdt [1], Robert Pietzcker [1], Adrian Odenweller [1,2], Anne Merfort [1,2], Renato Rodrigues [1], Jessica Strefler [1], Fabrice Lécuyer [1] & Gunnar Luderer [1,2]

The EU climate neutrality goal requires a strong reduction in fossil fuel use. However, whether a complete phase-out is feasible and desirable remains unclear. Using an integrated assessment model, we quantify the additional effort needed to achieve a nearly complete EU-wide phase-out of fossil fuels by 2050 compared to a least-cost net-zero scenario. We show that in the least-cost scenario fossil fuel consumption already decreases by 90% from 2020 to 2050 and is compensated by renewable power, direct electrification, as well as some hydrogen and biofuels. However, hard-to-replace oil-based hydrocarbons and natural gas persist primarily in the chemical industry, aviation and shipping. Phasing them out requires the large-scale deployment of costly carbon-neutral e-fuels, which substantially increases marginal abatement costs from 460 EUR to 630 EUR $tCO_2^{-1}$ (500-1000 EUR $tCO_2^{-1}$). Our works shows the additional transformation challenges if the EU aims to strengthen its climate policy commitment with a full fossil phase-out target.

About three-quarters of current greenhouse gas (GHG) emissions in the European Union (EU) result from the combustion of fossil fuels[1]. Therefore, fossil fuel use needs to decline substantially to achieve the 2050 climate neutrality goal legislated by the European Climate Law[2]. Ahead of the Conference of the Parties (COP) climate summit in 2023, the EU outlined its ambition for the world "to be predominantly free of fossil fuels well ahead of 2050"[3]. However, this ambition has not translated into concrete EU targets for reducing or completely phasing out fossil fuel consumption.

Residual fossil fuel use can be consistent with the EU's net-zero goal if the resulting carbon dioxide ($CO_2$) emissions are avoided through carbon capture and storage (CCS) at fossil installations, or if non-captured fossil $CO_2$ is compensated by carbon dioxide removal (CDR). However, the energy sector competes with other emitting sectors, like agriculture and industrial processes, for both CCS or CDR compensation. Moreover, the scale at which CCS and CDR will be available remains uncertain due to challenges related to upscaling, monitoring, reporting and verification, reversibility, finance, and sustainability impacts[4–6]. Integrated assessment modeling (IAM) scenarios

often overlook these aspects by assuming unrealistically rapid CCS deployment[7,8] that may lead to an overestimation of admissible residual fossil fuel use at the point of net-zero[9].

This poses the risk of creating false expectations as energy-sector stakeholders may scale down mitigation ambition in anticipation of sufficient future CCS or CDR to abate their fossil emissions. Without clear and realistic expectations about which sectors will be eligible for residual emissions at the point of net-zero, there is the risk that actors collectively slow down the transformation and hold back structural innovations in anticipation of CDR options to compensate their emissions. This underscores the importance of establishing clear eligibility criteria for residual emissions[10–12].

A possible solution to reduce this risk is to complement the EU 2050 net-zero target with a target on the full phase-out of fossil fuels that would rule out residual emissions in the energy sector. The appeal of such target mainly lies in creating an additional focal point for the EU energy transition that would signal increased political commitment, create clarity about climate-compatible investment decisions, and safeguard against political backlashes[13]. Additionally, a fossil

[1]Potsdam Institute for Climate Impact Research, Member of the Leibniz Association, Potsdam, Germany. [2]Global Energy Systems Analysis, Faculty of Process Science, Technische Universität Berlin, Berlin, Germany. ✉e-mail: felix.schreyer@pik-potsdam.de

phase-out target would support the EU's ambition to become independent of fossil fuel imports, a priority that has become even more pressing since the start of the Russo-Ukrainian War[14].

However, the economic and feasibility challenges of an EU fossil phase-out by 2050 are still unclear. Most EU net-zero scenario studies anticipate some residual fossil energy use by 2050, finding CDR compensation more cost-effective than eliminating the last 10–20% of energy emissions[15–18]. Global IAM studies have identified abatement limits predominantly in agriculture and partially in the energy sector[19–22], and have not specifically investigated a full fossil phase-out. In contrast, energy system models (ESMs) have analyzed 100% renewable energy systems in the EU, but differ substantially from IAMs by focusing on the hourly dynamics of renewable energy supply without addressing real-world transformation inertias, demand side mitigation dynamics, as well as macroeconomic implications[23–25]. While ESMs have demonstrated the technical feasibility of a 100% renewable energy system, a plausible full-system pathway to a complete fossil phase-out remains unclear.

In this work, we explore the transformation dynamics, challenges and feasibility of a full fossil phase-out in EU net-zero scenarios up to 2050 using the IAM Regional Model of Investment and Development (REMIND). We find that a near-zero fossil EU energy system can be reached in 2050, but results in increasing marginal costs of up to 630 euro per ton of $CO_2$ (EUR $tCO_2^{-1}$) driven by a substantial demand for expensive e-fuels to substitute residual fossils primarily in chemicals, aviation and shipping.

## Results

### Transformation to a fossil-free EU energy system

We explore a range of EU net-zero scenarios up to 2050 with different levels of residual fossil energy (Supplementary Table 1). We do this by varying the maximum availability of $CO_2$ storage injection capacity that influences the allowed amount of residual fossil energy use at the point of net-zero. First, we model a least-cost net-zero scenario (*LeastCost-NZ*) where $CO_2$ storage injection is free to scale up to 2 Gt $yr^{-1}$ in the EU. This is our benchmark net-zero scenario without a fossil phase-out. Second, we model a fossil-free net-zero scenario (*FosFree-NZ*) where $CO_2$ storage injection is limited to 110 mega tons $CO_2$ per year (Mt $CO_2$ $yr^{-1}$), which is lowest level at which net-zero can be achieved with the model. To explore the solution space around these two core scenarios, we run intermediate scenarios with gradual changes in allowed $CO_2$ injection rates (*Intm1-NZ*, *Intm2-NZ*, *Intm3-NZ*). We also examine sensitivity scenarios in several dimensions (see Methods and Supplementary Table 2): Regarding technology development, we run scenarios with higher costs of green hydrogen (*ExpensiveH2*), lower costs of direct air capture (*CheapDAC*), increased $CO_2$ capture rates (*HighCaptureRate*), as well as increased technology scale-up costs (*HighScaleUpCost*). In standard scenarios, we assume that half of EU e-fuel demand is met by imports. Here, we investigate sensitivities with low-cost e-fuel imports at a fixed price of 150 euro per megawatt hour (EUR $MWh^{-1}$) (*HighImport*) and no e-fuel imports at all (*NoImport*). Finally, we vary land system assumptions running scenarios with high biomass availability (*HighBiomass*) and scenarios with an increased EU land carbon sink (*HighLandSink*).

All EU net-zero scenarios investigated in this study are characterized by a strong reduction of fossil fuel use. In comparison to the 1.5–2 °C scenarios for the European Region from the database of the Assessment Report 6 (AR6) of the Intergovernmental Panel on Climate Change (IPCC)[26], our EU net-zero scenarios explore a range of considerably deeper fossil phase-out trajectories, including a scenario with a nearly complete phase-out by 2050. While as of 2020 the EU depends on fossil fuels for about 75% of its primary energy, this share drops in our scenarios to about 60% by 2030 and to less than 1% by 2050 in the *FosFree-NZ* scenario, and 11% by 2050 in the *LeastCost-NZ* scenario (Fig. 1a). These results stand in stark contrast to the majority of 1.5–2 °C

scenarios from the AR6 database, where fossil fuels account for 19–57% (10th–90th percentile range, used throughout this paper for AR6 data) of primary energy in 2050 in the European region (Fig. 1a, gray lines). Similarly, EU-focused transformation scenarios from the European Climate and Energy Modeling Forum (ECEMF) also indicate residual fossil energy use in 2050 (Fig. 1a, blue lines and dots). There appears to be a model fingerprint as those previous REMIND scenarios from ECEMF tended to be already at lower levels of residual fossil energy demand than scenarios from other models.

Looking at a broad set of indicators in 2050, the deep fossil fuel reduction in our EU net-zero scenarios is enabled by an accelerated energy transition until mid-century, going beyond most 1.5°C scenarios in the AR6 database (Fig. 1b–k). Direct electrification of energy end-use in buildings, road transport, and industrial process heat is more pronounced as the electricity share in final energy of our scenarios increases to close to 60% in 2050 (Fig. 1g) compared to a median of 47% in the AR6 scenarios. Our scenarios also show higher variable renewable energy (VRE) shares of up to 83% in the electricity mix (Fig. 1e) that constitute the backbone of an emissions-free power system by mid-century. As a consequence of increased direct electrification, the share of hydrocarbon energy carriers in final energy decreases substantially (from 75% in 2020 to 27% in 2050, Fig. 1f). This leads to efficiency improvements driven by electrification that result in an overall reduction of total final energy demand by about 40% relative to 2020 (Supplementary Fig. 1), which is less pronounced in AR6 scenarios. Another notable difference is the supply of green hydrogen and e-fuels to replace fossil fuels in hard-to-electrify sectors (Fig. 1h). While almost non-existent in AR6 scenarios, their share reaches 9% in the *LeastCost-NZ* and 19% in the *FosFree-NZ* scenario. Generally, ECEMF scenarios are closer to the scenarios of this study, but still show notable differences with respect to the penetration of VRE power, electrification and hydrogen, particularly for scenarios from models other than REMIND. Through a combination of all the above levers, the *FosFree-NZ* scenario achieves a nearly complete phase-out of fossil fuels by 2050. It reaches this state without relying on large-scale bioenergy production (Fig. 1c, i) and achieves EU net-zero GHG emissions by 2050 at considerably lower CCS and CDR deployment in comparison to most AR6 and ECEMF scenarios (Fig. 1j, k).

Moving from the current fossil-based system in 2020 to the 2050 state, energy flows fundamentally change in our scenarios as renewable power becomes the backbone of a fossil-free energy system (Fig. 2). Relative to 2020, fossil primary energy consumption decreases by about 90% in 2050 in *LeastCost-NZ*, and even by 99.5% in *FosFree-NZ*, thus reaching a nearly fossil-free system. Fossil fuels are largely substituted by (non-biomass) renewable electricity, which is either used directly or indirectly via electricity-based fuels. By moving away from the combustion of hydrocarbons, this transformation comes with considerable efficiency gains in terms of total primary energy consumption (incl. renewables used for imported e-fuels), which decreases by around 30% in 2050 relative to 2020.

The transformation to a fossil-free EU energy system can be characterized by three different steps: decarbonization of the power sector, direct electrification of end-use sectors and de-fossilization of residual combustible fuels (Fig. 3). These steps tend to occur consecutively in time and at increasing marginal cost of abatement. First, at relatively low marginal cost of abatement, the power sector already transitions until the mid-2030s to a system based on VRE from wind and solar power (Fig. 3d and Supplementary Fig. 2). This allows for a phase-out of coal power and in turn a strong reduction of coal consumption (Fig. 3a and Supplementary Fig. 3). Second, direct electrification of final energy across end-use sectors unfolds up to 2040 (Fig. 3e) as electricity replaces mainly natural gas and oil for energy use in low-temperature heating and road transport. The consumption of natural gas and oil decreases less rapidly than for coal, though (Fig. 3b, c). Finally, the last and most expensive step of the energy transition is a

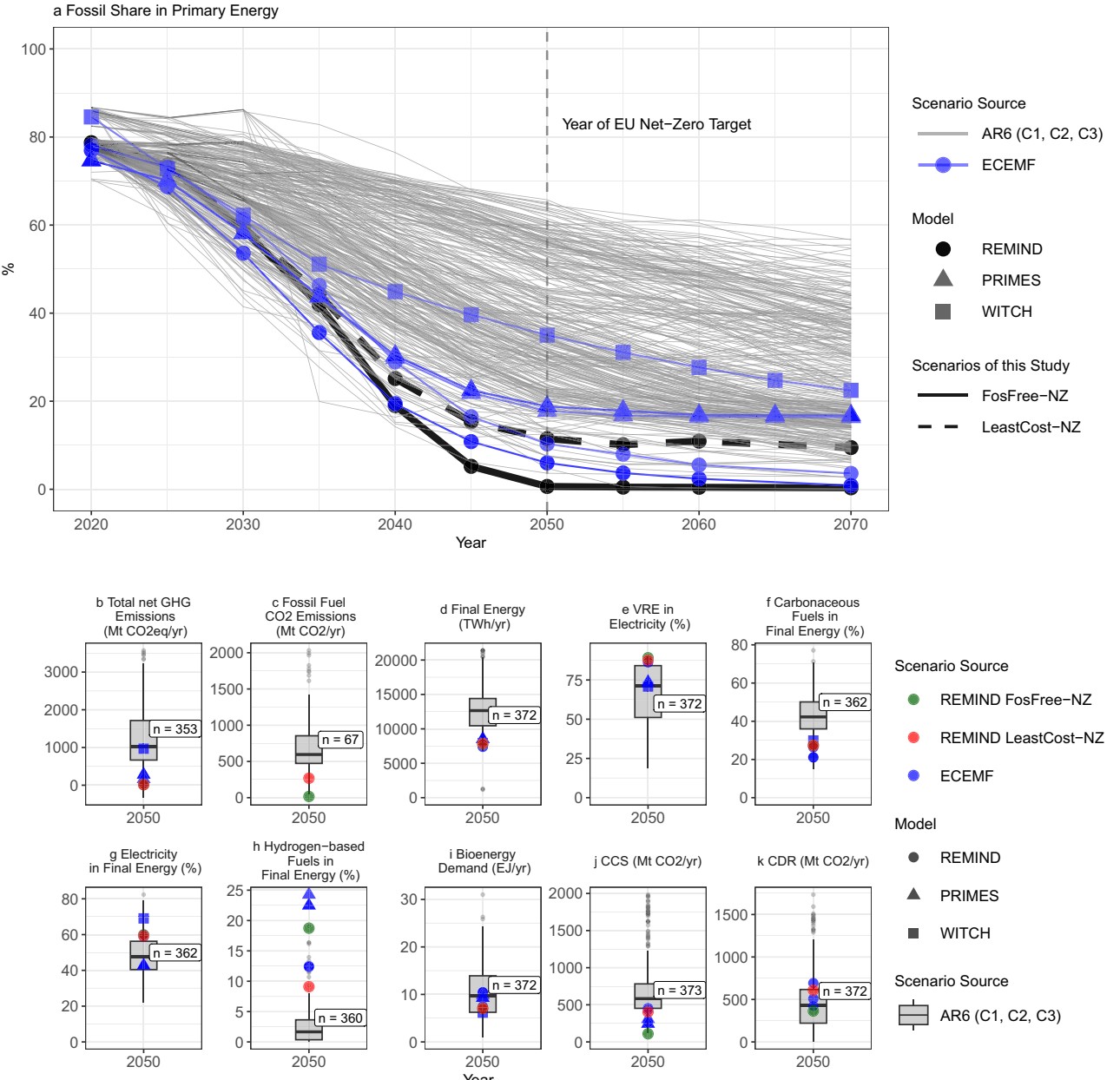

**Fig. 1 | Comparison of key scenario indicators in LeastCost-NZ and FosFree-NZ to scenarios from the literature. a** Share of fossil fuels in primary energy: Gray lines refer to scenarios from the database of the Intergovernmental Panel on Climate Change (IPCC) for the Sixth Assessment Report (AR6) and blue lines to scenarios from the European Climate and Energy Modeling Forum (ECEMF). Dot shapes of ECEMF scenarios refer to different models. EU scenarios of this study exhibit lower shares of fossil fuels than previously published scenarios. **b–k:** Boxplots and scatter plots with relevant mitigation indicators in 2050 for *LeastCost-NZ* and *FosFree-NZ* (red and green dots), AR6 scenarios (gray boxplots) and ECEMF scenarios (small blue dots). Dot shapes of ECEMF scenarios refer to

different models. The value *n* next to the boxplots gives the number of AR6 scenarios that report this indicator. The upper and lower hinges of the boxplots show 25th and 75th percentiles and the whiskers extend to the largest value within 1.5 times the distance between the 25th and 75th percentiles. Data beyond the whiskers are plotted as individual dots. The scenarios of this study and the ECEMF scenarios represent the EU region, while the AR6 data represent the larger European Region including non-EU countries. The AR6 scenarios were filtered for the climate mitigation categories C1, C2, and C3, representing 1.5–2 °C scenarios[26]. Overall, 9 scenarios from ECEMF project were included[67]. See details on comparison scenario in the Methods section.

switch to low-carbon fuels (based on biomass or green hydrogen) to avoid emissions in hard-to-electrify sectors that require combustible or carbonaceous fuels (Fig. 3f and Supplementary Fig. 4). This full de-fossilization of residual hydrocarbons in the last 10-15 years is the main difference between the *LeastCost-NZ* and the *FosFree-NZ* in the energy sector, which show similar levels of total hydrocarbon demand as the *LeastCost-NZ* already deploys most of the available electrification options until 2050 (Fig. 3f and Supplementary Fig. 5). To reach net-zero in the *LeastCost-NZ* scenario with residual fossil energy, there is

more CDR via increased usage of bioenergy with CCS (Supplementary Figs. 6 and 7).

## Residual fossils in 2050

In the *LeastCost-NZ* scenario, the fossil fuel carbon consumption (FFCC), defined as the carbon content of all fossil fuels consumed in the energy system, drops to 300 $MtCO_2$ $yr^{-1}$ by 2050 (Fig. 4a). The FFCC is equivalent to the sum of all energy-related fossil $CO_2$ emissions (excluding $CO_2$ process emissions e.g. from cement calcination) and

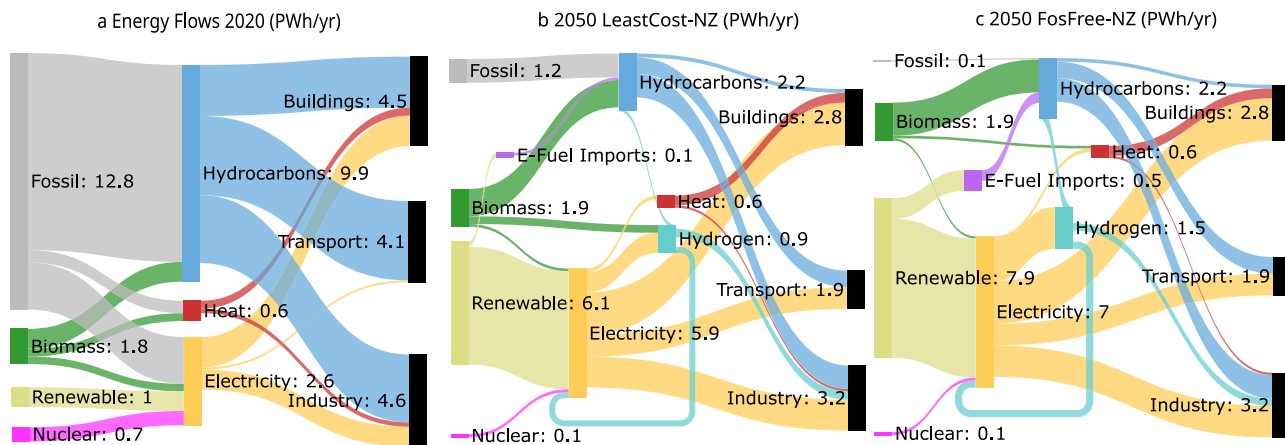

**Fig. 2 | Energy flows from primary to secondary to final energy for the EU in 2020 and 2050. a** EU energy flows in 2020 (modeled). **b** EU energy flows in the *LeastCost-NZ* scenario in 2050. In this scenario, residual fossil primary energy use of 1.2 PWh/yr is still present at net-zero. **c** EU energy flows in the *FosFree-NZ* scenario in 2050. This scenario reaches near-zero fossil primary energy use at net-zero as residual fossil fuels are replaced by carbon-neutral e-fuels. Numbers correspond to the total outflow of energy for primary energy (fossil, biomass, renewable, nuclear) and secondary energy (hydrocarbons, heat, electricity, hydrogen, e-fuel imports) in PWh yr⁻¹. Numbers correspond to total inflow for final energy (buildings, industry, transport) in PWh/yr. Flows amounting to less than 0.1 PWh yr⁻¹ have been removed for clarity.

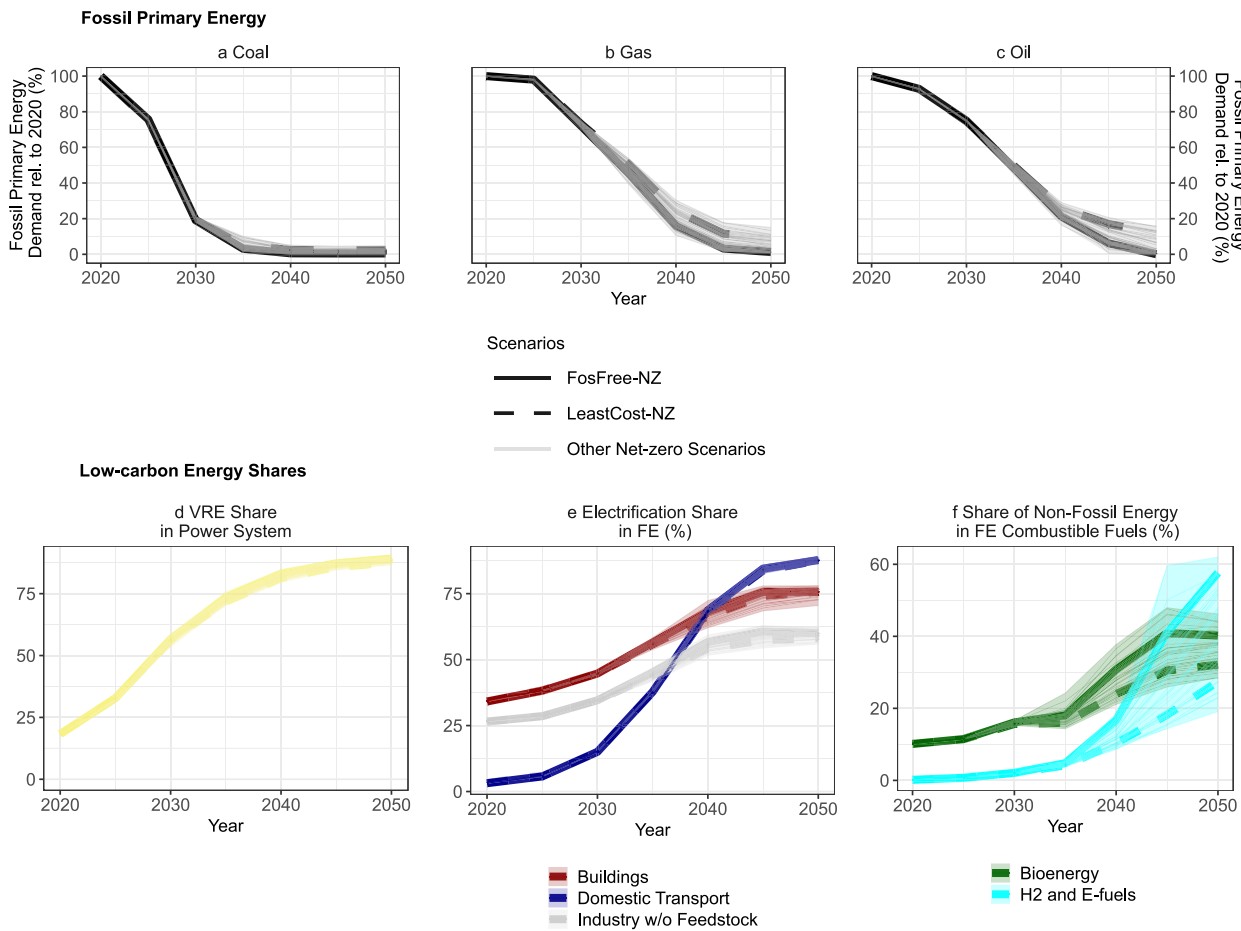

**Fig. 3 | Evolution of fossil energy demand and low-carbon energy shares over time.** Bold lines represent *FosFree-NZ* scenario (solid line) and *LeastCost-NZ* (dashed line). Thin lines represent all other net-zero scenarios of this study (see Supplementary Table 1 and Methods). The top panels show fossil energy demand: **a** EU demand for primary energy coal relative to 2020 levels. **b** As in **a** but for primary energy natural gas. **c** As in **a** but for primary energy oil. The bottom panels show clean energy shares: **d** Share of variable renewable energy (VRE), that is, wind and solar power in total power generation. **e** Share of electricity in final energy (FE) in buildings (red), industry without feedstocks (gray) and domestic transport (blue). **f** Share of non-fossil combustible fuels in final energy (FE) distinguished between biomass-based fuels (green) and hydrogen-based fuels, i.e., hydrogen and e-fuels (cyan). Fossil energy demand is drastically reduced in the net-zero scenarios as the power sector transitions to renewable energy, large parts of energy end-uses are electrified and residual fuel demands are supplied by carbon-neutral fuels based on biomass and green hydrogen.

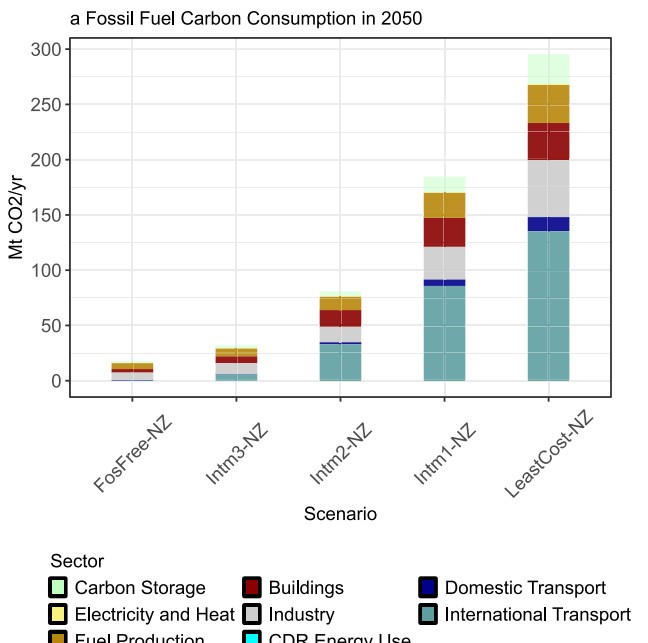

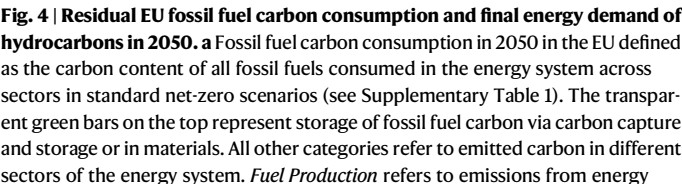

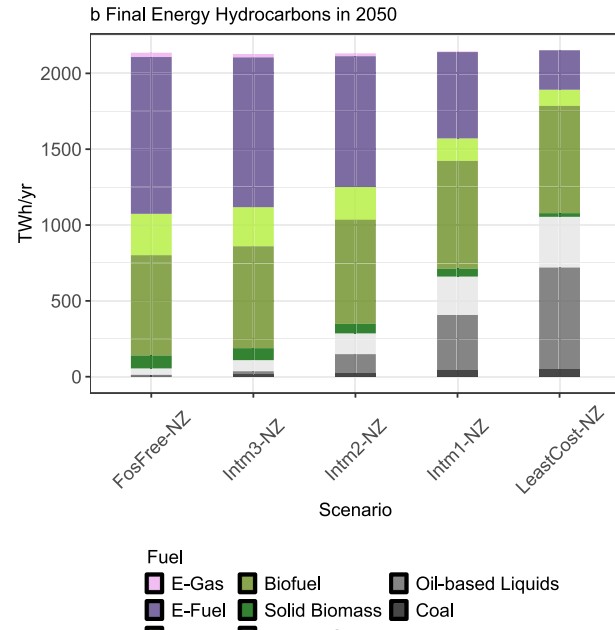

**Fig. 4 | Residual EU fossil fuel carbon consumption and final energy demand of hydrocarbons in 2050. a** Fossil fuel carbon consumption in 2050 in the EU defined as the carbon content of all fossil fuels consumed in the energy system across sectors in standard net-zero scenarios (see Supplementary Table 1). The transparent green bars on the top represent storage of fossil fuel carbon via carbon capture and storage or in materials. All other categories refer to emitted carbon in different sectors of the energy system. *Fuel Production* refers to emissions from energy conversion technologies outside the electricity and heat sectors, such as oil refineries or emissions related to fossil extraction. *CDR Energy Use* refers to energy use of carbon dioxide removal (CDR) technologies. **b** Final energy demand for hydrocarbons (solid, liquid, gaseous carbonaceous fuels) in 2050 in the EU in standard net-zero scenarios. The *FosFree-NZ* scenario almost fully phases out the last 300 Mt $CO_2$ yr$^{-1}$ of fossil fuel carbon present in the *LeastCost-NZ* scenario by replacing liquid fossil fuels with carbon-neutral e-fuels.

fossil carbon stored by CCS or in durable materials. We use this metric to distinguish between emissions reductions from avoiding fossil fuel use in the first place and permanent storage of fossil carbon. In the *LeastCost-NZ* scenario, fossil carbon storage plays a small role as 30 $MtCO_2$ yr$^{-1}$ are stored and 270 $MtCO_2$ yr$^{-1}$ are emitted. In *FosFree-NZ*, the FFCC reaches very low levels of 17 $MtCO_2$ yr$^{-1}$, which is a reduction of 99.5% from the 2020 level. Most of this carbon is emitted as the role of fossil carbon storage diminishes in *FosFree-NZ* since almost all of the limited $CO_2$ storage injection capacity in this scenario is used for sequestering carbon from bioenergy to generate CDR.

Residual fossil $CO_2$ emissions in the *LeastCost-NZ* scenario mainly come from oil-based liquid fuels and natural gas used in the chemicals sector (as fuel and feedstock), in international shipping and aviation (oil-based liquid fuels), and to a smaller extent in buildings (remaining gas boilers) (Fig. 4a). In addition to the combustion emissions accounted in these sectors, there are also emissions from the production of fossil fuels (fuel production sector in Fig. 4a). The electricity, district heating, and domestic transport sector are almost completely fossil-free even in *LeastCost-NZ*, and the buildings sector largely reduces fossil emissions by more than 90% in 2050 relative to 2020.

Moving from the *LeastCost-NZ* towards the *FosFree-NZ* scenario, an additional supply-side transformation occurs as an increasing amount of carbon-neutral fuels replace residual oil-based liquid hydrocarbons and fossil gas (Fig. 4b). Primarily, e-fuels, i.e., synthetic liquid fuels or chemicals produced from electrolytic hydrogen (Supplementary Fig. 4) and non-fossil $CO_2$, gradually replace fossil liquids in chemicals, aviation, and shipping. As e-fuels are an expensive mitigation option, they are only economical at scale once $CO_2$ storage is not available for further emissions abatement, which aligns with previous research[27–29]. Biofuels are also deployed but their use is not expanded in the *FosFree-NZ* scenario as the overall bioenergy potential is limited.

While in the *LeastCost-NZ* some biomass is also used to produce hydrogen, this biomass is shifted to produce biogas in the *FosFree-NZ* scenario (Supplementary Fig. 8). This additional biogas replaces remaining fossil gas used in industry and buildings. Overall, this enables a near complete fossil phase-out across energy supply and energy end-use sectors with a fossil share of primary energy below 1% (see Methods for details on limitations of modeling near-zero fossil use).

Total carbon capture in the EU by 2050 decreases from about 440 $MtCO_2$ yr$^{-1}$ in the *LeastCost-NZ* scenario to 260 $MtCO_2$ yr$^{-1}$ in *FosFree-NZ*, induced by the limitations on $CO_2$ storage injection (Supplementary Fig. 9). Instead of injection into geological storage, captured carbon is increasingly used for e-fuel production. This carbon usage takes up more than half of the carbon captured in the *FosFree-NZ*. Most of this carbon is biogenic, as direct air capture (DAC) is not competitive by 2050 in any of the standard scenarios (and deployed only in the *cheapDAC* sensitivity scenarios, see Supplementary Fig. 9b). $CO_2$ storage injection does not reach the scenario-specific limit of 2000 $MtCO_2$ yr$^{-1}$ in *LeastCost-NZ*, as only about 400 $MtCO_2$ yr$^{-1}$ are captured. This is because a more rapid scale-up until 2050 becomes increasingly expensive due to cost penalties in the model on high technology growth rates. However, the lower scenario-specific limit of 110 $MtCO_2$ yr$^{-1}$ is reached in the *FosFree-NZ* scenario. This minimum level of deployment is required to generate sufficient CDR for achieving economy-wide GHG neutrality (Supplementary Fig. 6).

### Cost implications of an EU fossil phase-out

A complete fossil fuel phase-out incurs additional costs beyond those of reaching the 2050 EU net-zero target. Marginal abatement costs (MAC) in 2050 (as reported by $CO_2$ prices required for the model to reach net-zero) increase from 460 EUR t$CO_2^{-1}$ in the *LeastCost-NZ* scenario to 630 EUR t$CO_2^{-1}$ in the *FosFree-NZ* scenario (Fig. 5a). The

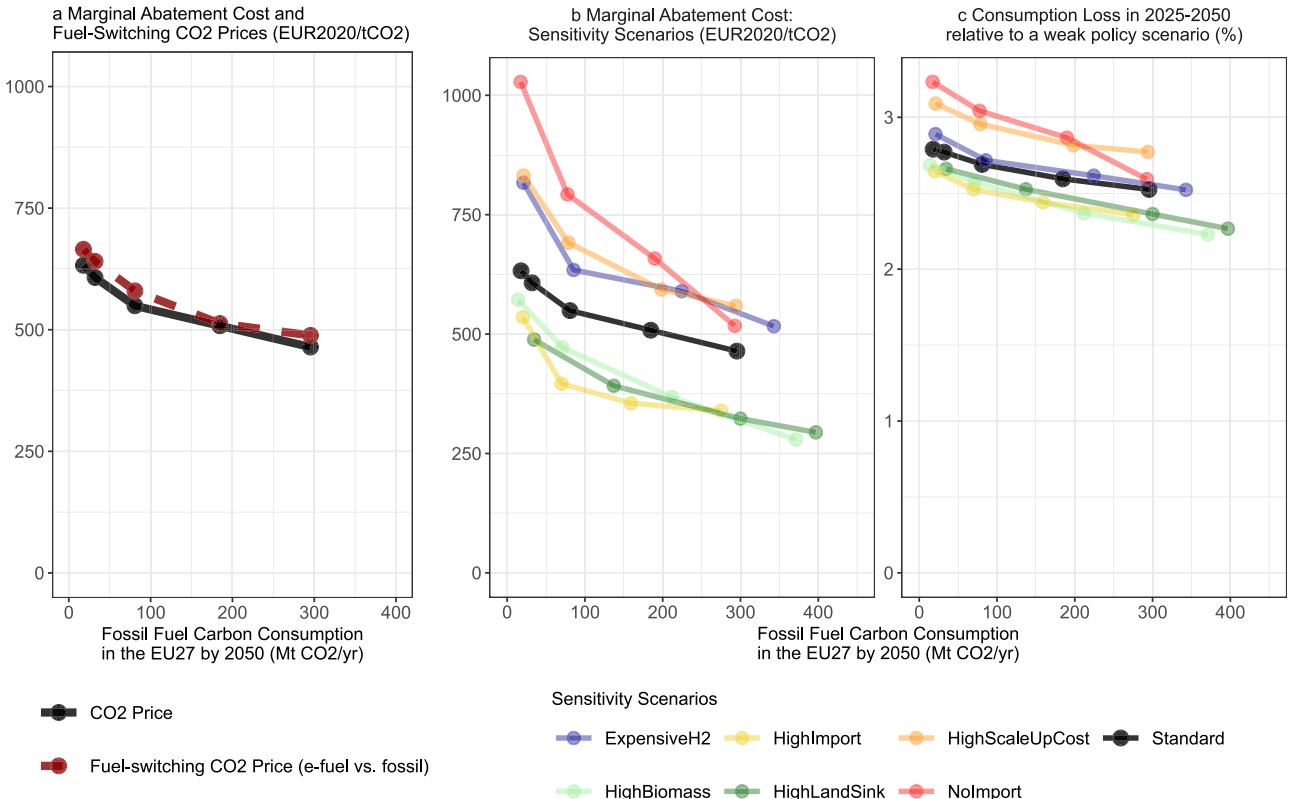

**Fig. 5 | Marginal abatement cost and aggregate consumption losses of EU net-zero scenarios with varying levels of residual fossil fuel consumption. a** EU marginal abatement cost on the y-axis against fossil fuel carbon consumption (carbon contained in total demand of fossil fuels) in 2050 on the *x*-axis across net-zero standard scenarios (black line). Moreover, corresponding fuel-switching $CO_2$ prices for replacing fossil liquid fuels by e-fuels based shadow prices from the model (dashed red line). **b** EU marginal abatement costs as in **a** across standard net- zero scenarios (black line) as well as sensitivity scenarios (thin colored lines). **c** EU aggregate 2025–2050 consumption losses (discounted at 3% per year) across standard net-zero scenarios (black line) as well as sensitivity scenarios (thin colored lines) relative to the *Weak Policy* scenario (see "Methods") on the *y*-axis. The *x*-axis shows again fossil fuel carbon consumption in 2050 as in the other panels. See "Methods" section for the scenario assumptions and the calculation of consumption losses.

MAC curve in this domain of low FFCC aligns well with the fuel-switching $CO_2$ price, which is the carbon price at which e-fuels are competitive with liquid fossil fuels based on the energy prices derived from our model (see dashed red line in Fig. 5a). This means that the MAC for the last about 300 MtCO$_2$ yr$^{-1}$ of FFCC are governed by the cost difference between e-fuels and fossil liquids (see also sensitivity scenarios in Supplementary Fig. 10). The increase in fuel-switching $CO_2$ prices between *LeastCost-NZ* and *FosFree-NZ* can be explained by increasing e-fuel cost as a result of increasing demand and decreasing cost of oil-based liquids as a result of decreasing oil demand (Supplementary Fig. 11).

At low FFCC, MAC are quite sensitive to scenario assumptions (Fig. 5b). In scenarios with higher biomass availability (*HighBiomass*) or abundant low-cost e-fuel imports at 150 EUR MWh$^{-1}$ (*HighImport*), MAC decrease in the *FosFree-NZ* scenario setting to 570 EUR tCO$_2^{-1}$ and 540 EUR tCO$_2^{-1}$, respectively. If the land sink from existing forests contributes more CDR (*HighLandSink*), climate neutrality could be reached with even more limited CCS. It leads to lower MAC at the same level of FFCC. This is because e-fuel production becomes cheaper since the competition with $CO_2$ storage over non-fossil captured carbon is reduced. Under the *LeastCost-NZ* CCS setting, the *HighBiomass and HighLandSink* sensitivities increase FFCC up to 400 MtCO$_2$ yr$^{-1}$ relative to 300 MtCO$_2$ yr$^{-1}$ in the standard *LeastCost-NZ* as more CDR is available. In contrast, under high cost of green hydrogen (*ExpensiveH2*) or without e-fuel imports (*NoImport*), MAC substantially increase to about 800 and 1000 EUR tCO$_2^{-1}$, respectively. Moreover, with additional

penalty costs on fast technology scale-up (*HighScaleUpCost*), MAC increases similarly to 800 EUR tCO$_2^{-1}$ since it becomes more costly for the model to reach the required level of e-fuel production by 2050. Higher $CO_2$ capture rates (*HighCaptureRate*) and variations on the cost of direct air capture (*CheapDAC*) hardly affect MAC in *FossFree-NZ*, and only slightly reduce them in *LeastCost-NZ* with high CCS availability (Supplementary Fig. 12). Generally, the MAC curves tend to follow a convex shape with stronger cost increases at low FFCC. Overall, the sensitivity analyses indicate that $CO_2$ prices required for a full fossil phase-out until mid-century strongly depend on the availability of low-cost carbon-neutral fuels.

Aggregate consumption losses over 2025–2050, which are representative of the total cost of climate change mitigation, are not as sensitive to our scenario assumptions as MAC in 2050. The relative changes in EU-wide consumption losses across scenarios (calculated relative to a weak climate policy scenario, see "Methods" section) are considerably smaller than the increase in MAC (Fig. 5c). This is plausible because only specific sectors that still demand liquid or gaseous hydrocarbons (chemicals and international transport) or hydrogen (industry) see an increase in energy prices in *FosFree-NZ* in comparison to *LeastCost-NZ* (Supplementary Fig. 13). In contrast, electricity prices that are relevant for the large segment of electrifiable sectors do not increase between the two scenarios. This indicates that the additional costs of a transformation to a full fossil phase-out are concentrated on specific energy carriers and sectors that make up a relatively small share of the total economy.

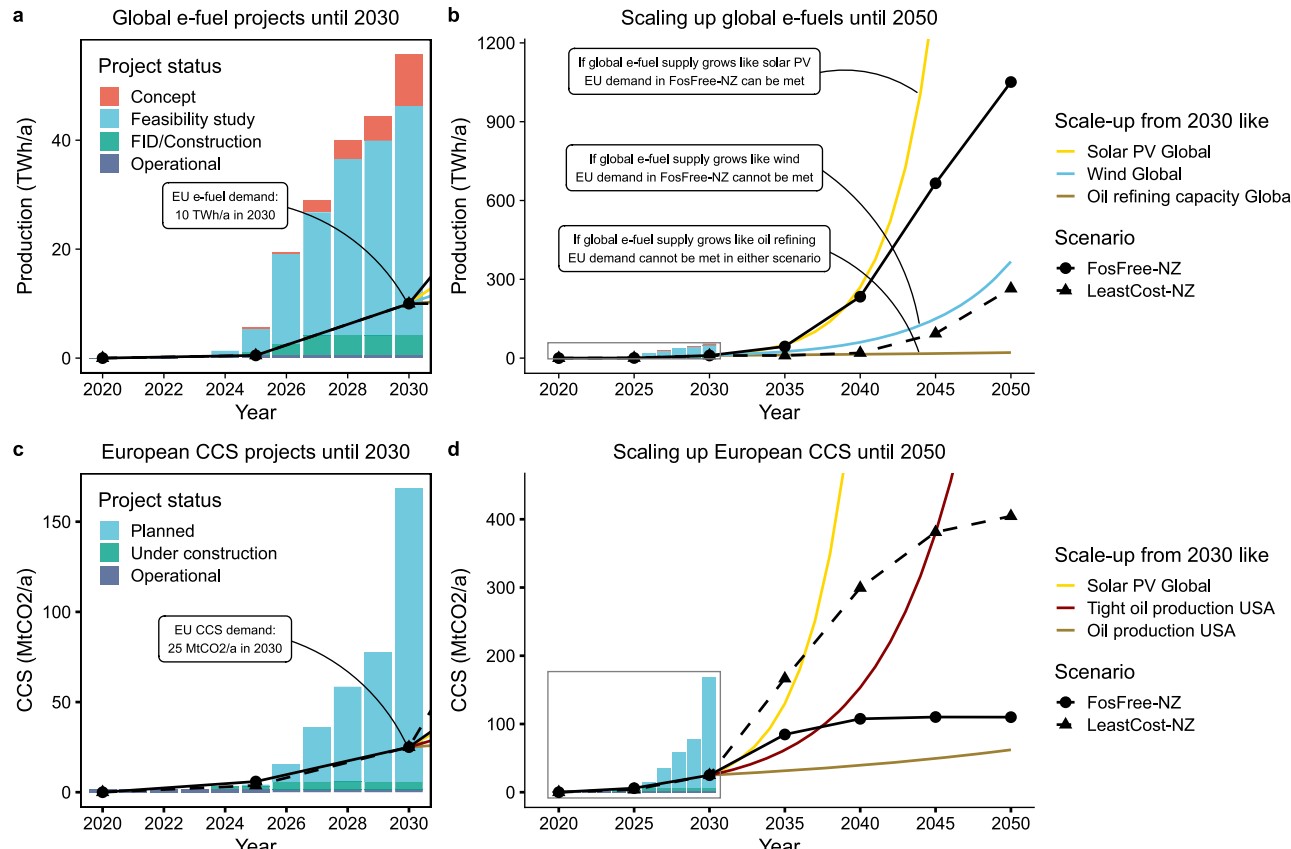

**Fig. 6 | Scenario-dependent trade-off between scaling-up global e-fuel production or European carbon capture and storage capacity. a** Global e-fuel project announcements by status. By 2030, we assume that 10 TWh yr$^{-1}$ of global e-fuel production capacity can be realized, corresponding to the aviation and maritime e-fuel sub-quotas legislated by the EU (see "Methods"). **b** Scaling-up of global e-fuels until 2050 following historical global 20-year growth rates of solar (39% per year) and wind power (20% per year) from 2003 to 2023, and oil refining capacity from 1965 to 1985 (5% per year, see "Methods"). Note that the depicted comparison of energy production shows the historical 20-year growth rate of the corresponding technologies starting from 10 TWh yr$^{-1}$ in 2030 and not the exact historical time series of these technologies. Global e-fuel production would need to grow as fast as solar photovoltaics (PV) and a large share would need to be sold to the EU in order to meet EU e-fuel demand in the *FosFree-NZ* scenario. **c** European carbon capture and storage (CCS) project announcements by status. By 2030, we assume that 25 MtCO$_2$ yr$^{-1}$ of European CCS capacity can be realized, corresponding to the 88% failure rate in Kazlou et al.[7] for the year 2030 (see "Methods"). **d** Scaling-up of European CCS until 2050 following historical global 20-year growth rates of global solar PV (39% per year), total US oil production from 2003 to 2023 (5% per year), as well as the 17-year growth rate of US tight oil production from 2007 to 2024 (20% per year, see "Methods"). European CCS capacity would need to grow almost as fast as US tight oil in order to meet the requirements in the *LeastCost-NZ* scenario, whereas the *FosFree-NZ* scenario keeps CCS at a minimum.

## Feasibility challenges of scaling up e-fuels or CCS

Reaching the EU net-zero target with or without residual fossil energy has implications, in particular, for the required deployment of e-fuels and CCS. EU e-fuel demand rises to more than 1000 terrawatt hours per year (TWh yr$^{-1}$) in 2050 in the *FosFree-NZ* scenario and European CCS capacity in the *LeastCost-NZ* scenario grows to about 400 MtCO$_2$ yr$^{-1}$. Given the nascent state of either technology, this poses substantial scale-up challenges.

For e-fuels, global project announcements add up to 56 TWh yr$^{-1}$ by 2030. Of these projects less than 1% are already operational and less than 7% have reached a final investment decision or are under construction (Fig. 6a). Global e-fuel production as required by the *FosFree-NZ* scenario could only supply EU e-fuel demand under two conditions: First, 10 TWh yr$^{-1}$ of production capacity needs to be realized by 2030 to supply the existing EU e-fuel quotas. Second, global e-fuel production would need to grow as rapidly between 2030 and 2050 as solar photovoltaics over the last 20 years and, moreover, the EU would need to secure a substantial share of the global market (Fig. 6b). Lower growth as experienced historically by e.g., wind power or oil refining, a technology more similar to e-fuel production, would not suffice. Given the complexity of the e-fuel supply chain, including green hydrogen

production and non-fossil carbon capture, these growth rates imply a crisis-like emergency deployment as investigated by previous feasibility studies on electrolysis or DAC[30,31].

For CCS, European project announcements are 168 MtCO$_2$ yr$^{-1}$ by 2030, of which only 1% are operational and less than 2.5% are under construction (Fig. 6c). Assuming a historical failure rate of 88% for announced projects[7], total European CCS capacity would reach 25 MtCO$_2$ yr$^{-1}$ by 2030, which would still be below the EU target of 50 MtCO$_2$ yr$^{-1}$[32]. Starting from 25 MtCO$_2$ yr$^{-1}$ in 2030, the *LeastCost-NZ* scenario would require CCS in Europe to grow up to 2050 almost as fast as US tight oil production over the last 17 years, while lower growth rates of e.g., total US oil production in the same historical period would not suffice (Fig. 6d).

There is a trade-off between high scale-up requirements for either CCS capacity or e-fuel production, depending on whether or not the EU net-zero target is to be reached with a full fossil phase-out. While long-term growth rates for CCS in the *LeastCost-NZ* scenario would not need to be as high as for e-fuels in the *FosFree-NZ* scenario, the scale-up of either technology can only be achieved with a short-term policy push up to 2030 as well as sustained long-term growth rates afterwards.

## Discussion

This study investigates the transformation dynamics and challenges of a full phase-out of fossil fuels in the EU, along with the climate neutrality goal by 2050. In summary, we find that in a least-cost net-zero scenario (*LeastCost-NZ*) without exogenous limits on CCS, fossil fuel consumption already decreases by 90% in 2050 relative to 2020 at MAC of 460 EUR $tCO_2^{-1}$. Residual fossil fuels are mainly oil-based liquids and some natural gas. They can be phased out with additional transformation efforts on the energy supply side by massively scaling up carbon-neutral e-fuels, which increases the MAC to 630 EUR $tCO_2^{-1}$. In such a full fossil phase-out scenario (*FosFree-NZ*), MAC are particularly sensitive to assumptions about bioenergy availability, e-fuel imports, and technology scalability (500–1000 EUR $tCO_2^{-1}$ range). In comparison, Victoria et al.[25] find MAC in 2050 of around 300 EUR $tCO_2^{-1}$ in scenarios with ambitious EU climate targets and very fast technology up-scaling. In REMIND, we model more realistic up-scaling as technology costs increase non-linearly with increasing growth rates. In ECEMF scenarios (see Fig. 1), EU carbon prices also tend to be around 600 EUR $tCO_2^{-1}$ in 2050, yet in scenarios with higher residual fossil fuel use compared to our *FosFree-NZ* scenario. Despite steeply increasing MAC to replace the last 10% of fossil fuels, aggregate economic costs do not increase as much in our scenarios since high MAC are only incurred in specific sectors with a small share in the total economy. This suggests that transformational challenges may be managed more easily if concerns about the distribution of abatement cost are addressed.

Our IAM study goes beyond existing scenario literature by exploring a complete phase-out of fossil fuels by 2050 in the EU. Scenarios from the AR6 database typically show substantially higher fossil fuel use by mid-century and beyond. First, diverging scenario designs can explain this difference: unlike AR6 scenarios compatible with 1.5–2 °C global warming, our study imposes net-zero GHG emissions in the EU and exogenous limits to CCS. Second, each model has its own characteristics, and REMIND tends to be particularly reactive to climate policies in terms of emissions abatement[33], featuring high renewable power and electrification shares and limited use of fossil fuels and CCS[34,35]. Methodologically, it is important to understand that the full fossil-phase-out is triggered by an endogenous response to increasing carbon prices in the model as a result of limiting the availability of CCS. This contrasts with studies investigating global scenarios with low residual fossil emissions as a result of a broader portfolio of mitigation options (lifestyle changes, advanced technologies) relative to a standard mitigation scenario where these options are not available[19,20].

Following previous EU net-zero scenario studies, our results confirm the importance of renewable power expansion, electrification, carbon-neutral fuels, and CCS[15,18,36]. While previously the role of e-fuels was mostly highlighted in scenarios with less pronounced demand-side electrification[18,37–39], we show here that phasing out fossil fuels requires the simultaneous achievement of extensive electrification and drastic e-fuel scale-up. This enables the EU to reach climate neutrality with limited reliance on $CO_2$ storage (around 100 $MtCO_2$ $yr^{-1}$), an important aspect given that the spatial distribution of biogenic $CO_2$ sources and prospective geological sinks may not allow for more than 200 $MtCO_2$ $yr^{-1}$ by 2050[40].

The comparison to literature underscores several limitations of our study. First, we do not specifically look at demand reduction measures that affect residual hydrocarbon use e.g., by circularity in the chemicals sector or lifestyle changes like reduced air travel[41–43]. Second, our model ignores technological options to switch away from hydrocarbon fuels in aviation and shipping. This assumption is nonetheless plausible considering the limited timeframe until 2050 and the nascent nature of these technologies[42,44]. Third, our scenarios assume that the EU land carbon sink from existing forests sustains its current absorption of 240 $MtCO_2$ $yr^{-1}$, which is within the range of 100–400 $MtCO_2$ $yr^{-1}$ estimated by Pilli et al.[45]. Substantial uncertainty surrounds future forests and relates to management practices as well as climate change impacts[45,46]. Moreover, land carbon removals are not permanent, which makes it problematic to fully account them as offsets for fossil $CO_2$ emissions[47]. Fourth, our model aggregates the 27 EU countries into just 8 regions, which restricts our insight into the geographical aspects of the energy transition. While we do not expect fundamental changes in the observed dynamics, a higher spatial resolution will be needed to investigate national scenarios and cross-European energy grids[24,48,49].

Our results have policy implications with respect to technology development and target setting in the EU. First, expansion of renewable power and electrification can lead a long way to the phase-out of fossil fuels, but technologies like CCS or e-fuel production will be needed to abate the last 10% of fossil fuels. Broad electrification is a necessary precondition for an extensive (90%) or nearly complete (>99%) phase-out of fossil fuels by 2050, with an electricity share in final energy close to 60% in all our scenarios. It emphasizes the importance of a policy strategy that clearly prioritizes electrification and reserves green hydrogen, e-fuels, and CCS only for specific applications[7,36,39,50,51].

In addition, substantial carbon capture of more than 200 $MtCO_2$ $yr^{-1}$ is required in all net-zero scenarios, even as $CO_2$ storage is limited in *FosFree-NZ*. Investing in non-fossil carbon capture is therefore essential as it is needed for both e-fuel production and CDR via $CO_2$ storage. E-fuel production and $CO_2$ storage technologies should be developed in parallel to create robust pathways that secure against potential delays in upscaling one of them. Given the ambitious scale-up of these technologies in our scenarios and current project delays[7,52], the window for reaching an extensive or nearly complete fossil phase-out by 2050 is rapidly closing unless increasingly large projects can be realized in the next years.

Second, the decision on an EU fossil phase-out target by 2050 hinges on a fundamental trade-off involving different types of risks and uncertainties. On the one hand, such a target would create an additional focal point for the EU energy transition and avoid deterring mitigation due to false expectations about the compatibility of fossil fuels with net-zero. It would provide clear signals to investors about the course of the energy transition, and help establish political credibility, a key ingredient to a functioning EU emissions trading scheme (ETS)[13,53,54]. On the other hand, a full fossil phase-out comes with a considerable increase in marginal abatement costs. This is mainly driven by the need to rapidly scale-up e-fuel supply to 1000 TWh $yr^{-1}$ in 2050, which is about as much as the total current EU fossil fuel consumption in aviation and shipping.

One viable middle course could entail defining partial and sector-specific targets for a fossil phase-out. Following the scenario ranges derived in our study, the EU could define a target of 90–100% reductions in fossil energy by 2050 relative to 2020, along with the net-zero goal, and break this target down to the sectoral level by extending the existing framework of EU renewable energy targets[55]. Nearly complete phase-out targets could apply to the building sector, road transport, and industrial process heat, while residual fossil use may be allowed in aviation, shipping, and chemicals. The exact formulation and flexibility of such targets certainly calls for political negotiation, which could be a challenging process given their far-reaching implications and the divergent interests of stakeholders. Ultimately, this trade-off between stabilizing target expectations and maintaining market flexibility involves different layers of social, economic, and political aspects that go beyond the scope of our energy system modeling perspective.

Several open questions remain to be addressed in future research on low-fossil energy systems. First, given the supply-side challenges to scale up non-fossil hydrocarbons, it is important to better understand how this pressure can be alleviated by demand-side measures like circular economy and lifestyle changes. Second, our study does not

explore the dynamics around bioenergy provision and implications for land systems. Investigating the competition and trade-offs between e-fuel and bioenergy to supply non-fossil hydrocarbons would be of great value. Finally, the phase-out of fossil fuels was one of the key elements of the COP in Dubai in 2023. Spelling out global fossil phase-out scenarios that inform international climate negotiations will be particularly relevant in the run-up to the next global stocktake by 2028.

## Methods
### Model description
We use a version of the IAM REMIND based on the release version 3.3.1, which can be found on GitHub (https://github.com/fschreyer/remind/tree/FossilFree_master). REMIND is an energy-economy model that allows to investigate transformation pathways with respect to different regional or global climate targets[56]. It conducts an interpoal Ramsey-type welfare maximization including a detailed energy system as well as simplified representations of the mitigation of non-$CO_2$ emissions in land use and other sectors. We apply the model in a setting of 21 world regions, including 8 regions of the EU. We simultaneously enforce a global cumulative peak $CO_2$ budget of 650 Gt$CO_2$ from 2020 representing a 1.5 C scenario with low overshoot and an EU-wide target of net-zero GHG emissions in 2050.

REMIND resolves key mitigation options for a deep transformation of the energy system. It covers all energy demand across the end-use sectors buildings, industry, transport, and carbon dioxide removal, including carriers used for energy purposes as well as material use (feedstocks). The model features an industry sector divided into the subsectors steel, cement, chemicals, and other industry[57], a detailed representation of passenger and freight transport[58], and a comprehensive energy supply system including technologies to produce electricity-based hydrogen and hydrocarbons[34,39,59]. Novel technologies are subject either to endogenous learning (e.g., wind, solar, electrolysis, DAC) based on installed capacity (learning-by-doing) or exogenous cost projections over time. $CO_2$ can be captured in the model from different streams (energy supply, industry, DAC) and can be either stored geologically (CCS) or used for the production of synthetic liquid or gaseous fuels via carbon capture and usage (CCU). Moreover, the model features a parameterization of VRE integration and flexibility effects in a power system with fluctuating renewables. For a more detailed description, please see the above references on different aspects of the modeling system.

There are a number of modeling features that are particularly relevant for the results of this study. First, carbon prices in the target year (referred to as marginal abatement costs in the main text) and energy prices are an endogenous outcome of the model. The model adjusts carbon price trajectories across iterations until all regional and global climate targets are met. The 2050 EU carbon price can therefore be interpreted as the marginal abatement cost to reach net-zero emissions in 2050. Energy prices are numerically computed shadow prices of energy balance equations in the model (secondary energy balance and final energy balance). They represent the monetary gain the model sees for having an additional unit of the respective energy carrier at free disposal.

Second, the representation of green hydrogen production has been improved relative to previous model versions. Hydrogen can be produced from different energy carriers, including electricity, biomass, natural gas, and coal. However, for electrolytic hydrogen, i.e., green hydrogen, there is the challenge of capturing the option of flexible electrolysis operation in a power system with fluctuating renewables. As REMIND only matches supply and demand in 5-year time steps, the dynamics of a VRE-based power system have to be parameterized based on results from hourly power system models[59]. To parameterize the flexible operation of electrolysis, we used data from the power system model Enertile for high-VRE scenarios of Germany[60] to derive a relation between the share of electrolysis in

annual electricity demand and the average electricity price for electrolysis normalized by the annual average electricity price. We used this relation to reduce the electricity price that electrolysis sees in REMIND, depending on the share of electrolysis in total power demand. In our standard net-zero scenarios, this results in a range of about 20 (*LeastCost-NZ*) to 35 EUR MWh$^{-1}$ (*FosFree-NZ*) average EU electricity price for electrolysis in 2050.

Third, we represent the option of producing synthetic (liquid) fuels and synthetic gas via two technologies that can convert hydrogen and $CO_2$ to hydrocarbon liquids or gas, respectively. Liquid synthetic fuel production is parameterized based on the Fischer-Tropsch synthesis technology, including subsequent cracking of hydrocarbons, while the synthetic gas technology represents a methanation process. For liquid fuel production, we do not differentiate between different energy carriers (gasoline, diesel, kerosene, naphtha, etc.) but implicitly assume that the process is optimized for a selectivity of hydrocarbons that fits the liquid fuel demand profile of REMIND. In the text, we refer to the outputs of the technologies as (liquid) e-fuel and e-gas as a simplification although the hydrogen used in these synthetic production processes may originate to a small share also from other technologies than electrolysis (see Supplementary Fig. 4). Finally, we consider the combustion of these synthetic fuels as carbon-neutral and account their emissions with the sector that captured the $CO_2$ to feed into the process. This is in line with accounting regulation of the EU emissions trading system (ETS), which only treats permanent storage of $CO_2$ as emissions abatement (not CCU) and combustion of CCU-based synthetic fuels as carbon-neutral[61,62].

### Scenario assumptions
We distinguish three different categories of net-zero scenarios in our analysis (Supplementary Table 1). First, *FosFree-NZ* and *LeastCost-NZ* are the two core scenarios to compare pathways to net-zero with minimal and maximal residual fossil energy in our analysis. Second, the standard scenarios include the two core scenarios as well as the scenarios with intermediate residual fossil energy use (*Intm1-NZ, Intm2-NZ, Intm3-NZ*) that only differ in terms of their allowed annual $CO_2$ storage injection rate. Finally, the sensitivity scenarios comprise all scenarios with modifications of other scenario assumptions (other than CCS injection). That is, they vary biomass availability (*High-Biomass*), the cost of direct air capture (*CheapDAC*), the cost of electrolytic hydrogen (*ExpensiveH2*), the availability of e-fuel imports to the EU (*HighImports* and *NoImports*), the technology scale-up cost (*HighScaleUpCost*), the $CO_2$ capture rate of biomass gasification (*HighCaptureRate*), and the size of the land sink (*HighLandSink*). In the following, the assumptions of all scenario types are explained.

There are several assumptions that are common to all of our EU net-zero scenarios. First, all of them reach the EU climate goals for 2030 and 2050 by assumption. For the 2030 target, we use a target of 2120 Mt$CO_2$ yr$^{-1}$ total GHG emissions in the EU, including emissions from international transport (bunker fuels) within the EU, representing -55% emissions reduction relative to 1990 values. For 2050, we use a target of net-zero GHG emissions in the EU, including all emissions from international transport.

Second, in standard net-zero scenarios, we limit EU bioenergy production in the standard scenarios at 7.5 Exajoules (EJ) yr$^{-1}$ in line with the low scenario of Ruiz et al.[63]. Moreover, we suppress bioenergy imports from other world regions by charging a high bioenergy import tax such that bioenergy consumption in these scenarios is strictly limited. This was done to respect sustainability criteria for the production of bioenergy in and outside the EU. It leads to a bioenergy consumption in the range of the 1.5LIFE and 1.5LIFE-LB scenarios developed by the EU Commission[16].

Third, we assume in all net-zero scenarios that sectoral and infrastructure policies enable a widespread electrification of energy end-uses. Our scenarios generally assume that socio-technological

barriers to the adoption of battery electric vehicles in road transport, as well as heat pumps in industry and buildings, such as infrastructure availability, first-mover disadvantages, and status quo biases, are overcome with increasing usage and that they can compete with existing fossil technologies based on their economics.

Fourth, in the standard net-zero scenarios we assume that 50% of the EU e-fuel demand by 2050 will be imported at a fix price of 150 EUR MWh$^{-1}$. This is to reflect that not all of the e-fuel demand will need to be met by domestic supply. The e-fuel demand is endogenous to the model. However, e-fuel imports are exogenous to the optimization process such that the e-fuel prices in the model reflect the marginal cost of producing another unit domestically.

The assumption we vary to explore pathways with different levels of residual fossil energy at net-zero is the CCS injection rate (Supplementary Table 1). Specifically, we consider different limits on the maximum annual $CO_2$ injection into geological storage within the EU of 110 MtCO$_2$ yr$^{-1}$ (*FosFree-NZ*), 130 MtCO$_2$ yr$^{-1}$ (*Intm3-NZ*), 180 MtCO$_2$ yr$^{-1}$ (*Intm2-NZ*), 350 MtCO$_2$ yr$^{-1}$ (*Intm1-NZ*), and 2 GtCO$_2$ yr$^{-1}$ (*LeastCost-NZ*).

In addition to the dimension of CCS, we vary other parameters in the sensitivity scenarios. The scenario assumptions behind the sensitivity scenarios are summarized in Supplementary Table 2. The *HighBiomass* case increases the EU bioenergy production potential to 12.5 EJ yr$^{-1}$ following the reference scenario by Ruiz et al.[63], which is higher than all of the scenarios from the EU Commission[16]. The *CheapDAC* scenario makes very optimistic assumptions about the cost development of DAC. We set the learning rate from 15 to 25% per year, reduce the energy demand per unit of captured carbon by 50% and assume that at least 100 MtCO$_2$ yr$^{-1}$ DAC capacity are operating globally by 2040, which drives down the capital costs via learning-by-doing. This increased learning rate is higher than what is typically indicated by technology assessments[64].

The *ExpensiveH2* scenarios assume a higher floor cost for electrolysis (300 \$ kW(el)$^{-1}$ instead of 100 \$ kW(el)$^{-1}$) and a smaller benefit from flexible operation that results in range of about 30–50 EUR MWh$^{-1}$ average EU electricity prices for electrolysis between the *LeastCost-NZ* and *FosFree-NZ* scenarios in 2050. Electrolysis is still assumed to run flexibly at a capacity factor of 0.38 as in the standard scenarios. This sensitivity represents a case in which electrolysis can access less low-price hours in the power system due to less price variability in the system and increased competition with other flexible electricity demand.

Moreover, we run two scenario variants with different assumptions of the availability of e-fuel imports. In the *HighImport* scenarios, we provide the model with the possibility to import an unlimited amount of carbon-neutral e-fuels to the EU regions at a fixed price of 150 EUR MWh$^{-1}$. These imports are not sourced from other model regions but can be accessed by the respective region via a separate import stream. The price level was chosen in consideration of technoeconomic literature on the cost of importing electricity-based Fischer-Tropsch fuels to the EU[65,66]. We also run a scenario variant, *NoImport*, without e-fuel imports, where the EU has to supply all of its e-fuel demand on its own.

In addition, we run a *HighScaleUpCost* sensitivity scenario where we increase the default cost markup on technology capital cost that penalizes fast ramp-up rates. This means that the CAPEX of technologies will increase more at high levels of technology growth relative to the standard case. This serves to explore the impact of technology scale-up barriers on our scenario results. We also run a *HighCaptureRate* scenario that increases the $CO_2$ capture of biomass gasification (in the supply chain to either produce biogenic H$_2$ or bio-based Fischer-Tropsch fuels) from 90% (default case) to 99% to explore the sensitivity of our results.

Finally, we run a set of sensitivity scenarios with a higher land sink and slightly adjusted assumptions about CCS injection rates (*HighLandSink*). The land sink in 2050 is increased to 370 MtCO$_2$ yr$^{-1}$,

which is between the official EU 2030 land sink target of 310 MtCO$_2$ yr$^{-1}$ and the maximum potential of 400 MtCO$_2$ yr$^{-1}$ in 2050 estimated by Pilli et al.[45]. In the *FossFree-NZ-HighLandSink* and the *Intm2-NZ-HighLandSink* scenarios, the maximum CCS injection rate is lowered relative to the *FossFree-NZ* (10 MtCO$_2$ yr$^{-1}$ instead of 110 MtCO$_2$ yr$^{-1}$) and the *Intm2-NZ* scenario (110 MtCO$_2$ yr$^{-1}$ instead of 180 MtCO$_2$ yr$^{-1}$). This is because the scenarios should be comparable in terms of residual fossil energy consumption by 2050. As more CDR is contributed from the land sink, less CDR is necessary from CCS to reach the EU net-zero GHG goal in 2050.

## Scenario comparison data

We take scenario data from the AR6 scenario database[26] and the ECEMF database[67] for comparison of our results. We describe the main filtering steps of the data in the following. For further details, please see the script used for our analysis, that is appended in our data repository.

We filter the AR6 scenarios for categories C1, C2, and C3 representing scenarios that are consistent either with 1.5 or "well-below" 2 °C global temperature stabilization as they are the most ambitious recent IAM mitigation scenarios available. We filter the data for scenario results of the European Region (R10EUROPE). Note that his has a wider regional scope than the EU27 region of our scenarios generated in this study, including non-EU European countries, which is why we focus on comparing share values instead of absolute values. Next, we exclude some scenarios and models based on sanity checks of the scenario data. We exclude (1) scenarios with share data shown in Fig. 1 that exceed 100% for any time step. We also exclude (2) scenarios with carbon capture and storage exceeding 2 GtCO$_2$ yr$^{-1}$ in the European Region by 2050, which is clearly outside of feasible limits. We exclude (3) scenarios with negative gross energy-related emissions in any time step. This leaves us with 382 scenarios. However, not all of them are included in each indicator of Fig. 1, as some models do not report all of the variables used for calculating the indicators.

Regarding the ECEMF data, we filter for the EU27 region and the scenarios *DIAG-C400-lin (version: 1)*, *DIAG-C400-lin* and *WP1 NetZero [Pre-Release 2023-11-14] (version: 1)*. These are the most ambitious scenarios currently in the database as the first two scenarios see a linear carbon price increase to 400 EUR tCO$_2$$^{-1}$ and the last scenario achieves the net-zero EU goal by assumption. This leaves us with 9 scenarios from four models: REMIND 2.1, PRIMES 2022, WITCH 5.1, Euro-Calliope 2.0, where the net-zero scenario is only provided by the first two models. Our boxplots of ECEMF data therefore rely on a much smaller set of scenarios than boxplots of AR6 scenarios (see Fig. 1).

## Calculation of fuel-switching $CO_2$ price

The average EU fuel-switching $CO_2$ prices between liquid e-fuels and fossil liquids shown in Fig. 5 are calculated by a weighted aggregation of the regional price differences between both energy carriers in the following way:

$$\text{FSCP}_{\text{EU}} = \sum_r \frac{\text{FE}_{\text{fossil},r}}{\sum_k \text{FE}_{\text{fossil},k}} * (P_{\text{efuel},r} - P_{\text{fossil},r}) * \frac{1}{\text{Emifac}} \tag{1}$$

In Eq. (1), FSCP$_{\text{EU}}$ represents the fuel-switching $CO_2$ price between liquid e-fuels and fossil liquids in the EU. FE$_{\text{efuel},r}$ represents the final energy e-fuel demand by region $r$. $P_{\text{fossil},r}$ and $P_{\text{efuel},r}$ represent the final energy prices of fossil liquids and liquid e-fuels before taxes respectively (including transmission and distribution cost). Emifac denotes the $CO_2$ emissions factor for the combustion of liquid fuels in our model which is 0.26 tCO$_2$ MWh$^{-1}$. The price difference is weighted by the regional share of fossil liquids demand in all EU regions to give most weight to those regions which still have fossil liquids demand in this scenario. This is done to give less weight to regions, which already reach very low or no residual fossil liquids demand in a scenario. Here, e-fuels obviously cannot substitute fossil liquids anymore and shadow

prices of fossil liquids in the model are not meaningful as they become zero or negative and would thus distort the metric.

## Weak policy scenario and consumption losses

For the calculation of aggregate consumption losses in the net-zero scenarios, we moreover run a *Weak Policy* scenario, which reflects a scenario with continued low climate policy ambition in the EU, similar to the situation in 2020 before the European Green Deal policy framework was legislated. The EU $CO_2$ price in this scenario is not adjusted to meet a specific EU emissions target but set exogenously to remain from 2020 on at 30 EUR $tCO_2^{-1}$ throughout the modeling time period. Moreover, sector-specific technology policies like the sales ban of internal combustion engine passenger cars from 2035[68] or subsidies to support heat pump uptake implemented in the net-zero scenarios are not applied. This *Weak Policy* scenario does not reach the EU climate targets. It reduces total GHG emissions to only about 2200 $MtCO_2$ $yr^{-1}$ by 2050, which is the level of 55% reductions relative to 1990 that is supposed to be reached already in 2030 according to the legislated EU climate targets.

This scenario serves as a reference point to calculate the aggregate economic consumption losses in the net-zero scenarios as a measure of the total cost of the transformation. The consumption losses, also referred to as total mitigation cost, are calculated as the difference in time-discounted aggregated consumption of the net-zero scenarios and the Weak Policy scenario. That is,

$$\text{Consumption Losses} = \frac{\left[\sum_{t=2025}^{2050} C_{t,wp} * e^{-r^*(t-t_0)} - \sum_{t=2025}^{2050} C_{t,nz} * e^{-r^*(t-t_0)}\right]}{\sum_{t=2025}^{2050} C_{t,wp} * e^{-r^*(t-t_0)}} * 100 \quad (2)$$

Here, $C_{t,wp}$ refers to the total consumption in the EU in time step $t$ in the Weak Policy scenario. $C_{t,nz}$ refers to the total consumption in the EU in time step $t$ in the net-zero scenario for which the consumption losses are calculated. $e^{-r^*(t-t_0)}$ is the discount factor and we choose a discount rate of $r = 0.03$ in line with common practice of calculating consumption losses in IAM analyses[69, p. 361]. The time step $t_0$ is 2025, the first time step of the aggregation period.

## Upscaling analysis

For e-fuels, we use global project announcements from the IEA Hydrogen Production Projects Database[70], last published in October 2024, including projects that report either *Synfuels* or *MeOH* (Methanol) as the product. For projects that report *Various* as the product, we distribute the capacity in equal shares to each given end-use product, and finally include the corresponding project share for the *Synfuels* end-use product. We only include projects with an announced launch year and derive approximate e-fuel production levels from the announced electrolysis capacity (in terms of electrical input) by using an electrolysis efficiency of 70%, an electrolysis capacity factor of 38%, and an e-fuel synthesis efficiency of 70%. We exclude projects in the *Other/Unknown* category and distribute projects in the *DEMO* category to *Operational* if the announced launch year is in the past or *FID/Construction* if the announced launch year is in the future. For 2030, we assume that 10 TWh $yr^{-1}$ of global e-fuel production capacity can be realized, which approximately corresponds to the quantities required for the EU e-fuel sub-quotas implemented within the ReFuelEU Aviation and FuelEU Maritime regulations[71].

For CCS, we use project announcements from the IEA CCUS Projects Database[72], last published in March 2024, including projects for EU27 plus Norway and the United Kingdom, as the EU may export $CO_2$ for storage to these countries in the long run. We only include the project types *Full chain*, *Storage*, and *T&S* (transport and storage) and projects that report *Dedicated storage* as the fate of carbon. For the project capacity, we use the column *Estimated capacity by IEA*. For 2030, we assume that 25 $MtCO_2$ $yr^{-1}$ of European CCS capacity can be realized, which corresponds to applying the recent failure rate of 88%

for CCS projects[7] to all uncertain project announcements in the *Planned* category.

For the growth rate comparison, we obtain global compound annual growth rates (CAGRs) of wind, solar photovoltaics and hydro power production for the last 20 years (2003–2023), using the EI Statistical Review of World Energy 2024[73]. We also use a historical 20-year CAGR of total US oil production (2003–2023) and a 17-year CAGR of US tight oil production (2007–2024) from this dataset. This serves to as comparison for the CCS scale-up requirements in our scenarios as exploration, installation, and operation of $CO_2$ storage and oil production rely on similar technological infrastructure and processes. We chose US tight oil growth as an optimistic reference point for future CCS scale-up because its rapid growth in particular around 2010 has been one of the most dynamic market developments in fossil fuel extraction in recent history.

## Limitations on the fossil phase-out

The *FosFree-NZ* scenario still features a very small amount of fossil energy use in the EU by 2050 (65 TWh $yr^{-1}$ representing 0.5% of total primary energy) as well as fossil fuel $CO_2$ emissions (16 $MtCO_2$ $yr^{-1}$, representing 0.5% of current levels). We do not investigate lower shares of fossils as this scenario captures the main dynamics of the fossil phase-out and model results with even lower fossil shares are difficult to interpret. This is mainly because marginal abatement cost represented by $CO_2$ prices are highly sensitive when we do model runs with even lower 2050 fossil fuel consumption. For very small quantities the model does not see fuel-switching abatement options anymore and $CO_2$ prices strongly increase as the model has to retort to very expensive mitigation options such as further reducing energy service demand. When reducing the availability of CCS even further, there is a point at which model runs do not converge anymore, indicating that the given EU climate target of 2050 net-zero GHG emissions cannot be reached in this setting.

## Data availability

The data used and generated in this study have been deposited on Zenodo under: https://zenodo.org/records/17132530. For the scenario comparison data, we used the AR6 and ECEMF scenario explorers[26,67]. For e-fuel and CCS projects, we retrieved data from the IEA Hydrogen Projects Database[70] and the IEA CCUS Database[72]. For calculating historical growth rates, we used data from the Energy Institute[73].

## Code availability

The REMIND code used to generate the results can be found on GitHub[74]: https://github.com/fschreyer/remind/tree/FossilFree_master. The data analysis scripts and plotting data can be found at https://zenodo.org/records/17132530.

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

## Acknowledgements

This research has received funding from the German Federal Ministry of Education and Research under grant agreement 03SFK5A0-2 (Ariadne; F.S., F.U., A.O., R.P.) and the European Union's Horizon 2020 Research and Innovation Programme under grant agreements 101022622 (ECEMF; R.R.), 101081604 (PRISMA; G.L.), 101183367 (NEWPATHWAYS; F.L.), and grant agreement No. 101081521 (UPTAKE; A.M.). We thank F. Benke for providing support with the visualizations. Moreover, we thank the research software engineering group of the REMIND team for technical support and the IT team at PIK for providing the high-performance cluster computing environment used for the generation of the results.

## Author contributions

F.S., F.U. and G.L. conceptualized the research. F.S. conducted the scenario modeling analysis and A.O. conducted the upscaling analysis. F.S., A.O. and F.L. created visualizations. F.S., R.P., F.U., A.O., A.M., F.L., and G.L. validated and interpreted the modeling results. F.S., F.U., R.P., A.O., A.M., R.R., J.S. and G.L. contributed to the concepts and implementation of the modeling system. F.S. wrote the manuscript with contributions from A.O., F.U., R.P., A.M., J.S., F.L. and G.L. F.U., R.P. and G.L. contributed to the acquisition of funding used for this project.

## Funding

## Competing interests

The authors declare no competing interests.
