## [Transparent Peer Review file · Nature Communications]

From net-zero to zero-fossil in transforming the EU energy system

Corresponding Author: Mr Felix Schreyer

Version 0:

Reviewer comments:

Reviewer #1

(Remarks to the Author)

Dear Author(s),

Thank you for your well-written and structured manuscript.

This is a significant study with a comprehensive methodology and meaningful analysis of a zero-fossil scenario for the EU, which is novel and highly relevant to Nature Communications.

The evaluation of options for hard-to-decarbonise sectors is particularly insightful, where key uncertainties in DAC, H₂, and import prices accounted for in the sensitivity testing. Additionally, the presentation of the feasibility challenges in achieving such e-fuel implementation, such as in Figure 6 is impactful.

Although the methodology is sound and well laid out, with good supporting files, I do identify an issue. In 2050, the least cost scenario suddenly sets the land carbon sink to 370 MtCO₂/yr, while the fossil zero scenario maintains 240 MtCO₂/yr. This is not justified or discussed, or how such a number would be practically realised, and seems like an inconsistent input. Provided a 130 MtCO₂/yr value is substantial, implying rapid and huge transformations in EU's natural capital, likely impacting the REMIND solution space for that scenario, I would like much stronger justification, evidence, and/or sensitivity testing here – this needs careful consideration.

The discussion is another weakness of the paper, although it can be swiftly improved. Most of it is simply summarising the results, which is not necessary. It only lightly discusses limitations, not covering some major aspects and assumptions. Discussion of the results compared to previous works is highly encouraged to help distinguish the results and solidify novelty. Lastly, I think it's a missed opportunity not to include some clear recommendations for decision-makers and policy.

I believe this is important and substantial work, but I highly encourage improving the discussion to meaningfully communicate the key messages, clear implications and recommendations, and strengthen novelty.

I have attached detailed comments with further suggestions. I look forward to receiving your responses and revised article.

Best wishes,

(Remarks on code availability)

It is well-laid out and provides the required files - however, this is subject to a user's access to the REMIND IAM. This is standard.

Reviewer #2

(Remarks to the Author)

This is an excellent and very timely paper assessing fossil fuel phase-out in Europe. As far as I can tell the methodology is

solid and well-explained. The results are plausible. The only suggestion for revisions I have is to at least briefly discuss the policy implications of this work in more detail and be more specific about the "so what?" question.

(Remarks on code availability)

Reviewer #3

(Remarks to the Author)

The manuscript deals with an interesting topic that is consistent with the purpose of the journal. The manuscript has a suitable content for a scientific paper. The structure of the paper is good. The Data and Methodology section is also very well written in sufficient detail. The Results are also presented in a straightforward and clear way. The discussion section is good.

The manuscript has potential to be published but specific issues need to be addressed before publication.

1. In the abstract, authors must present the method used and the period of the study.
2. The authors must present in the final part of the study, the limits of research and future directions of research.

(Remarks on code availability)

The manuscript deals with an interesting topic that is consistent with the purpose of the journal. The manuscript has a suitable content for a scientific paper. The structure of the paper is good. The Data and Methodology section is also very well written in sufficient detail. The Results are also presented in a straightforward and clear way. The discussion section is good.

The manuscript has potential to be published but specific issues need to be addressed before publication.

1. In the abstract, authors must present the method used and the period of the study.
2. The authors must present in the final part of the study, the limits of research and future directions of research.

Reviewer #4

(Remarks to the Author)

This paper uses the REWIND IAM to model scenarios to achieve the EU 2050 economy-wide decarbonization goal. The authors construct a least-cost scenario, in which some fossil fuel is used, that serves as a baseline against which other scenarios are judged. This least-cost path achieves net-zero in part by employing CCS, CDR, and some carbon-neutral fuels and e-fuels. The objective of the paper is to determine a scenario that achieves net-zero by phasing out these remaining 10-15% of fossil fuels. The resulting fossil-free scenario has an abatement cost that is twice as much as that for the least-cost scenario and is differentiated from the other scenario by two important characteristics: greater e-fuels utilization and less CCS deployment. The authors then look at the projected growth rate for these technologies in reference to the historical growth rates of energy sources. This comparison demonstrates that full deployment of each technology by 2050 may be difficult to achieve. The authors conclude that while a goal of no fossil fuel may be a useful focal point for decarbonization, a better course may be to allow fossil fuel quotas for specific sectors to add planning certainty and relieve pressure on the power sector.

This paper is a thoughtful and well written piece on a pertinent topic that is also discussed in many papers from different perspectives. The comparison of the model outputs illustrates the differences in energy and technology reliance and is something that can be leaned on more in the discussion. While the premise of the paper concerns policy decisions, no concrete policy recommendations are made. There seem to be opportunities for this regarding aspects such as DACS, importing e-fuels, and biomass. If these cannot be discussed here, I look forward to reading the next paper that does so. This reviewer finds that the article can be published in Nature Communications after some minor revisions.

Questions and recommendations for the authors

1. In the abstract, the authors emphasize that one strength of the complete phase-out of fossil fuels is that it shows EU commitment to the climate target. The weakness is that it also requires transformation challenges. In the conclusion, the authors end with e-fuels can take pressure off CCS deployment, which is suggesting that some combination of both should be done to avoid the difficulties of large-scale e-fuel and CCS deployment and make it more likely that the target can be achieved. Why does the abstract not mention the least-cost CCS option and the strength of both approaches? Word count is not a sufficient answer.
2. Fig 1 should have same y-axes when possible.
3. In Figure 1, do the authors' scenarios directly compare to any of the AR6 scenarios, other than fuel use? What are the similarities? Is there a lesson from the common characteristics like more electrification is better, more hydrogen or a contrast in CCS and CDR?
4. There are three ECEMF lines in Figure 1. How does the middle one compare to the authors' Leastcost-NZ and the ECEMF scenario, which is similar to the authors' scenario but allows more fossil fuel? Can the authors compare abatement costs? Here it would help to label the ECEMF lines and dots in Figure 1 for the ones most like Leastcost-NZ and FosFree-NZ.
5. For Figure 2, it is difficult to read some of the font because of the white highlighting and the font not being black. Can the authors use a white background for the font when it is over green or red flows? Even when magnified it is difficult.

6. Figure 3F should have the same y-axis as Figures 3D and 3E.
7. CCS is mentioned for storage in the introduction and is mentioned for utilization in the results when injection is not used. These two uses should be directly called out in the introduction or combined as CCUS.
8. What CCS capture rate is used in the model? Capture rates of 95-99% (deep CCS) are currently being funded and actively studied in the literature. While the impact on the Leastcost-NZ case may be low, it will make more bio-genic fuel available and lower overall emissions. A sensitivity run similar to cheap DACSS may be insightful to show that it too does not matter.
9. In Figure 4, the colors for the legend are washed-out. Please put a black border around the lighter colors so those of us with some color blindness can differentiate them.
10. In Figure 5, are the authors indicating that DAC cost does not matter, since it overlaps with the standard scenario? This also seems to be true for high imports and high biomass availability. Can the authors comment on this in the paper?
11. The comparison arguments for Figure 6 are unclear. The nascent nature of these technologies is a formidable issue. It would be helpful to have a reference to the appropriate methods section here to understand how the growth rate comparison is constructed. The note to see Methods in the figure for panel (a) seems to exclude the other panels and these panels would benefit from the direction.
12. In looking at the figure the first question that comes to mind is “Are the solar and wind growth rates based upon projected capacity and scaled to plot as done in Panel d, or are they based on projections that are derated for capacity factors?”. The idea is right, but the energy comparison seems wrong. A useful comparison to put it in context may be to oil refineries. How many e-fuel facilities of a given capacity are required each year to reach the energy demanded and how many refineries are built each year or what percentage of the total European or world refineries does this capacity represent? What capital expenditure does this represent? Also, are the authors indicating that it will be difficult for the Leastcost-NZ scenario to meet the required output because the projected energy growth rate is below that for wind? If so, I missed the important point.
13. While e-fuels can be imported as needed, the paper concerns primarily EU e-fuel, so a stronger argument can be made if Figure 6(a,b) only represent EU demand and supply.
14. These arguments are stronger for CCS for solar and wind and how building refineries may be more appropriate. I don't think oil production in the US is a good comparison. All of these technologies have issues with growth rates increasing because of subsidies, changes in government policy to increase scale, and demand-pull versus technology-push. If the authors want to bound the required growth rates, taking the historical technology growth rate and plotting it is fine, but label it as a 20% or 5% growth rate rather than the specific technology.
15. As a comparison, Hanna, Abdulla, Xu, and Victor have a paper in Nature Communications that describes the emergency deployment of DACS and compares the required build rate and money with historical examples. Solar is used in this paper but as a reference high growth rate that was also achieved with US Liberty ships in WWII. Hanna, R., Abdulla, A., Xu, Y., & Victor, D. G. (2021). Emergency deployment of direct air capture as a response to the climate crisis. Nature communications, 12(1), 368.
16. Is the 88% failure rate extrapolated to 2050 given learning rates are involved in the process? If so, this is unlikely. Yes, for nascent technology the first efforts may fail, and we have seen CCS facilities shutdown because the price support is no longer there (WA Parish) or the cost overruns are excessive (Kemper). However, these failure rates will decrease as the learning rate improves and causes of the failure are removed (policy, financial, and technical). If the failure rate is extrapolated, a discussion of the Pareto of failures would be helpful. Rubin, Herzog, Mac Dowell, Heuberger are good sources for learning rates.
17. The line colors in Figure 6 Panel d are hard to discern. Can the authors use bolder colors for the thin lines?
18. In line 329 the authors mention the pressure on CCS is relieved with more e-fuel. Do the authors ever recommend multiple approaches to achieve a higher likelihood of achieving net zero? This is done in the last paragraph and should be linked back to this line and thought.
19. Total energy in Ext Data Fig 1 does not show non-heating energy.
20. SI 4 should have the same y-axes.

Grammatical editing to improve the paper

1. Line 9 should read “Quantifies the additional effort...” instead of “Quantification of the additional effort...”
2. Line 11 should read “By 2050, least-cost net-zero scenario abates 87% of EU fossil fuel use with increased renewable power, electrification, and biofuels”. Last comma is missing after electrification in original if the authors don't use suggest change.
3. For the third bullet, the detail about the oil-based hydrocarbons is not required for the highlight. This bullet may be shortened to “Carbon-neutral e-fuels are crucial to replace the remaining 13% of hardest-to-abate fossil fuels used for international transport and chemicals”.
4. For line 16, the “(500-1000 €/tCO₂)” can be removed because this is a highlight, and the detail can be noted in the main text. Similarly, “roughly” is not required. The authors can also remove “a” prior to least-cost and replace it with “the” because the second bullet introduces this scenario.
5. Line 24 should have a comma after aviation.
6. In line 25, “about” can be omitted.
7. In line 26 the expect range can also be omitted, as can “by 2050”. The range is presented in the body and 2050 is already mentioned twice in the abstract.
8. In line 27, omitting “also” increases the tension and make the abstract more dramatic.
9. In line 34, replace “About” with “Almost” and hyphenate three-quarters.
10. While EU is a well-known acronym, I don't recall the journals policy for requiring to define acronyms like this or CO₂. Greenhouse gas emissions is defined. Same with COP28, etc.
11. In line 41 the authors introduce CCS and later on the authors mention captured carbon being used for e-fuels. Is there ever a distinction between CCS and CCUS (carbon capture for utilization and storage)?
12. In line 46, replace “fast” with “rapid”. Also, if the authors are including DACS in this thought, an appropriate reference for

this is Hanna, Abdulla, Xu, and Victor, “Emergency deployment of direct air capture as a response to the climate crisis.”

13. In line 46, remove the comma after “aspects” and replace it with “by”. To emphasize the overestimation, the comma in line 47 can be removed and “which” replaced with “that”. The overestimation is an important part that should be emphasized.
14. In line 50, “false” is a better word than “wrong”.
15. In line 50, it should be “energy-sector stakeholders scale-down mitigation ambitions in anticipation of sufficient CCS or CDR deployment to abate emissions.”
16. Line 57 should “is to” instead of “would be to”.
17. Line 58 should be “in order to” instead of “that would”.
18. Line 60 needs a comma after “decisions”.
19. Line 69 omit “also”.
20. In line 70 replace “they” with “these studies”.
21. Line 79 should read “, and derive abatement cost curves for the remaining 10-15% of fossil emissions...”.
22. In line 80 the authors can omit “EU” because the authors just said so but why do the authors change from sector to system? Sector is preferred for consistency.
23. In line 94, change to “lowest level at which net-zero can be achieved with the model...”. My assumption from the sentence is that this is a limitation of the model.
24. In line 97, break this up into two sentences. “We also examine sensitivity scenarios with high...”.
25. In line 98, add a comma after “(highBio)”.
26. In line 203 replace “they” with our EU NZ scenarios” or something like this to differentiate the authors work from the IPCC work.
27. In line 108 add “the” after “of”.
28. In line 114 add a comma after transport.
29. In line 115 start a new sentence with “This is done by increasing the share of renewable energy in the electricity energy mix to 60% by 2050...”.
30. In line 117, is 27-30% in 2050? The discussion on efficiency improvements should be a new sentence because it is long and complicated.
31. In line 119, “substitute” should be “replace”.
32. Line 144 should say “Fossil fuels are largely replaced with renewable-based electricity.” Directly and indirectly doesn’t add anything and biomass isn’t typically considered as renewable energy.
33. Line 157 should be “, and de-fossilization of...”.
34. In line 160, do the authors mean higher renewable penetration? If so, this phrase is better. (“...to one with high renewable penetration that reduces coal consumption.” If one needs to, one can also add “high renewable penetration of variable solar and wind capacity that...”)
35. In line 162 is natural gas and oil replacing coal, or is something replacing all three?
36. In line 164 the “most expensive step” seems lost. If the authors wish to emphasize the time-step and the expense-step, it can be added after “finally”. It seems unnecessary to have this clause though.
37. In line 189, can be simplified to “which is a reduction of 99.5% from the 2020 level.”
38. Line 196 should have a comma inserted “liquid fuels), and...”
39. Line 198 should have a comma inserted “district heating, and domestic transport sector...”
40. Line 199 should be broken into two sentences. “even in LeastCost-NZ. Furthermore, fossil emissions in the buildings sector are largely reduced...”
41. In line 202, “substitute” should be changed for “replace”.
42. In line 203 adding parentheses makes it easier to read. “Primarily, e-fuels (i.e. synthetic liquid fuels or chemicals produced from electrolytic hydrogen (Ext Data Fig 5) and non-fossil CO₂ (Ext Data Fig 6))...”
43. Line 205 needs a comma after “aviation”.
44. Line 208 is confusing. Do the authors mean “Bioenergy is used to produce hydrogen in the LeastCost-NZ scenario, and is used in the FosFree-NZ scenario to produce biogas that replaces fossil gas in industry and buildings (SI Figure 1).”
45. Line 219 should be rewritten. “Conventional CCS fails to reach the 2050 LeastCost-NZ scenario-specific limit of 2000 MtCO₂/yr, as less than 400 MtCO₂/yr is captured. However, the lower scenario-specific limit of 110 MtCO₂/yr is reached in the FosFree-NZ scenario.
46. “Fossil-fuel Carbon Consumption” should be hyphenated in line 227.
47. In line 245 marginal abatement costs can be replaced with MACs or “...MAC is quite...”
48. Line 246 should be “higher biomass availability...” and decrease should be decreases.
49. Since the authors defined MAC as singular, all uses of it should be singular—the verb needs an s.
50. Line 249 should read “...indicates that the CO₂ prices...”
51. Line 279 should be made into two sentences. The second sentence can begin “Of this required quantity, less than 1%...”
52. Line 281 should have a comma after 2030.
53. Line 282 is unclear and needs to be rewritten as two sentences.
54. There should be a comma after electrification in line 316.
55. In line 315, “in time” is not required after “consecutively” because it is implied.
56. Line 317 is too long and detailed. Please rewrite.
57. In line 317, “already quite impactful” is too informal. Perhaps “...are significant for achieving net-zero emissions: Decarbonization and electrification result in decreasing 2050 EU fossil primary energy consumption by 87% relative to 2020 levels at a marginal abatement cost of 300 EUR/tCO₂. This reduction is driven by a transition...”
58. When the authors add “...and some bio-based fuels” there needs to be a comma before and. Also, isn’t this the third step a contradiction to the statement that this covers only the first two steps? The rest of the sentence is more confusing.
59. Hyphenate fossil-fuel carbon consumption in line 319.
60. In line 320-321, at what point? “...moving from a least-cost to a fossil-free net-zero scenario requires that the focus changes from (cost solutions of some sort?) to supply-side energy solutions such as non-fossil hydrocarbons.”

61. In line 321, the secondary clause should be move to the beginning of the sentence.
62. In line 323 add “the” before “remaining”.
63. Hyphenate one-third in line 334. The journal style guide may differ.
64. In line 339, replace “but” with “and”.
65. In line 340, specify green hydrogen or make it clear in the results that the authors are only considering green hydrogen.
66. Line 346 should have a comma after “economy”. As much as what?
67. In line 350 the authors can emphasize fossil-free target to link to their scenario.
68. In line 351 there should be a comma after “fossil use”.
69. In line 353, the authors can emphasis the obstacles for CCS by removing the “which” and replacing it with “that”. Otherwise, they need to add a comma.
70. In line 357, the sentence will have greater emphasis if the authors change it to “...TWh/yr—ten times the current EU liquid fuel consumption in aviation and shipping.”
71. In line 357, “They” should be changed to “These e-fuels”.
72. Line 359 should read “...that only allow a certain percentage of sector-specific fossil fuel use...”.

(Remarks on code availability)

Version 1:

Reviewer comments:

Reviewer #1

(Remarks to the Author)

Dear Author(s),

Thank you for your revised manuscript.

I am happy to recommend your article for acceptance and publication, subject to having appropriately addressed editor and other reviewer comments.

You have made significant efforts to address my concerns surrounding the land carbon sink assumptions and the expanded discussion on study implications; in addition to clarifying and improving other minor points. I highly commend this important work and believe it can have a high impact in Nature Communications.

Warm regards,

(Remarks on code availability)

As before: It is well-laid out and provides the required files - however, this is subject to a users access to the REMIND IAM. This is standard.

Reviewer #2

(Remarks to the Author)

(Remarks on code availability)

Reviewer #4

(Remarks to the Author)

I accept the modifications to the original transcript and recommend that the revised manuscript for publication.

(Remarks on code availability)

Dear Referees,

thank you for your helpful feedback on our submission. Based on your comments and questions, we had some internal discussions and revised the document in several ways. Below, you find a summary of the main changes:

- 1.) We revised the scenario assumptions of the standard scenarios. Previously, we had varied the size of the EU land sink, the cost of CO₂ storage as well as the maximum annual injection rate into CO₂ storage by 2050. In the updated scenarios, we only vary one assumption, which is the maximum annual injection rate into CO₂ storage. There are several reasons that led us to reconsider this choice. First, there is a large uncertainty about the future development of the EU forest carbon sink and an increase of the sink comes with land use and management implications that we cannot represent in our modeling system. Our new precautionary assumption maintains the land carbon sink at current levels of 240 MtCO₂/yr across all standard scenarios. This more cautious treatment is justified by the literature highlighting problems of land carbon accounting, the impact of management practices and the temporary nature of those carbon removals (Carton et al., 2021; Pilli et al., 2022; Verkerk et al., 2022). Nevertheless, we included a set of sensitivity scenarios with the previously higher land sink of 370 MtCO₂/yr. The effect of this enhanced sink can therefore be seen in the analysis, but does not feature the core scenarios. Moreover, we found it important to only vary one assumption across scenarios to derive our marginal abatement cost curves in Figure 5. If we vary more than one assumption on CCS and CDR across scenarios, the shape of the marginal abatement cost curve is affected by the sequence of how we apply these assumptions (e.g. it matters whether we decrease CCS injection rate or land sink first to generate low-fossil scenarios). We therefore think that this new scenario design makes it easier to interpret our marginal abatement cost curves and applies more reasonable assumptions on CDR.
- 2.) This new scenario design led to slight changes mainly in our *LeastCost-NZ* scenario. As a result, reductions in fossil energy consumption by 2050 are a bit higher, that is, 90% by 2050 relative to 2020 levels (instead of 87% previously). The fossil share in primary energy decreased from 16% in 2050 to 11% in the new runs. In turn, there is a larger supply of renewable electricity to compensate for this and the marginal abatement cost by 2050 increase to 460 EUR/tCO₂.
- 3.) We broadened the set of sensitivity scenarios (see Figure 5 and Extended Data Figure 6). First, we added the *HighLandSink* set of sensitivity scenarios with the enhanced land sink of 370 MtCO₂/yr. It shows the case of our previous *LeastCost-NZ* scenario in case of high CCS assumptions. In turn, under this setting net-zero by 2050 can be reached with even lower CCS at higher fossil fuel consumption relative to our standard *FosFree-NZ* scenario. Second, we ran a *HighScaleUpCost* sensitivity scenario to highlight the sensitivity of the marginal abatement costs to our assumptions on technology scalability of the e-fuel supply chain. This shows that the abatement cost are affected by how expensive it will turn out to realize high sustained growth rates of these technologies. Third, we ran a *HighCR* scenario to show the impact of high CO₂ capture rates of biomass gasification technologies on our results. Achieving very high capture rates of 99% (instead of 90%) only slightly decreased the marginal abatement costs.

From net-zero to zero-fossil in transforming the EU energy system
Revision 1, Nature Communications

- 4.) We reworked and extended the discussion section as recommended by all reviewers. First, we substantially shortened the summary paragraph in this section. Second, we extended the discussion of our results in comparison to scenarios from the IPCC AR6 database as well as existing EU net-zero scenario studies. Finally, we present a more comprehensive discussion of policy implications. We discuss technology choices necessary to reach the EU net-zero target as well as the benefits and challenges of setting (flexible) fossil phase-out targets on at sectoral level on the way to climate neutrality.

Please find our point-by-point responses in blue print below. Citations from the manuscript are printed in orange.

Reviewer #1

Dear Author(s),

Thank you for your well-written and structured manuscript.

This is a significant study with a comprehensive methodology and meaningful analysis of a zero-fossil scenario for the EU, which is novel and highly relevant to Nature Communications.

The evaluation of options for hard-to-decarbonise sectors is particularly insightful, where key uncertainties in DAC, H₂, and Import prices accounted for in the sensitivity testing. Additionally, the presentation of the feasibility challenges in achieving such e-fuel implementation, such as in Figure 6 is impactful.

Thanks a lot. We are glad to hear that.

Although the methodology is sound and well laid out, with good supporting files, I do identify an issue. In 2050, the least cost scenario suddenly sets the land carbon sink to 370 MtCO₂/yr, while the fossil zero scenario maintains 240 MtCO₂/yr. This is not justified or discussed, or how such as number would be practically realised, and seems like an inconsistent input. Provided a 130 MtCO₂/yr value is substantial, implying rapid and huge transformations in EU's natural capital, likely impacting the REMIND solution space for that scenario, I would like much stronger justification, evidence, and/or sensitivity testing here – this needs careful consideration.

Thanks for this important point. We agree that this is a crucial aspect and the assumption needs to be chosen with care. Your comment led us to reconsider the land sink assumption as described in the overview above under point 1. Please find a detailed answer below your point 7.

The discussion is another weakness of the paper, although it can be swiftly improved. Most of it is simply summarizing the results, which is not necessary. It only lightly discusses limitations, not covering some major aspects and assumptions. Discussion of the results compared to previous works is highly encouraged to help distinguish the results and solidify novelty. Lastly, I think it's a missed opportunity not to include some clear recommendations for decision-makers and policy.

Thanks for the remark. Based on your comments and those by other reviewers we reformulated the discussion section. It now reads as follows:

This study investigates the transformation dynamics and challenges of a phase-out of fossil fuels in the EU along with the climate neutrality goal by 2050. In summary, we find that in a least-cost net-zero scenario (*LeastCost-NZ*) without exogenous limits on carbon capture and storage (CCS), fossil fuel consumption already decreases by 90% in 2050 relative to 2020 at MAC of 460 EUR/tCO₂. Residual

fossil fuels are mainly oil-based liquids and some natural gas. They can be phased-out with additional transformation efforts on the energy supply side by massively scaling up carbon-neutral e-fuels, which increases the MAC to 630 EUR/tCO₂. In such a full fossil phase-out scenario (*FosFree-NZ*), MAC are particularly sensitive to assumptions about bioenergy availability, e-fuel imports and technology scalability (500-1000 EUR/tCO₂ range). In comparison, Victoria et al. (2022) find MAC in 2050 of around 300 EUR/tCO₂ in scenarios with ambitious EU climate targets and extremely fast technology up-scaling. In REMIND, we model more realistic up-scaling as technology costs increase non-linearly with increasing growth rates. In ECEMF scenarios (see Figure 1), EU carbon prices also tend to be around 600 EUR/tCO₂ in 2050, yet in scenarios with higher residual fossil fuel use compared to our *FosFree-NZ* scenario. Despite steeply increasing MAC to replace the last 10% of fossil fuels, aggregate economic costs do not increase as much in our scenarios since high MAC are only incurred in specific sectors with a small share in the total economy. This suggests that transformational challenges may be managed more easily if concerns about the cost distribution are addressed.

Our IAM study goes beyond existing scenario literature by exploring a complete phase-out of fossil fuels by 2050 in the EU. Scenarios from the AR6 database typically show substantially higher fossil fuel use by mid-century and beyond. First, diverging scenario designs can explain this difference: Unlike AR6 scenarios compatible with 1.5-2°C global warming, our study imposes net-zero GHG emissions in the EU and exogenous limits to CCS. Second, each model has its own characteristics, and REMIND tends to be particularly reactive to climate policies in terms of emissions abatement (Dekker et al., 2023), featuring high renewable power and electrification shares and limited use of fossil fuels and CCS (Luderer et al., 2022; Soergel et al., 2021). Methodologically, it is important to understand that the full fossil-phase-out is triggered by an endogenous response to increasing carbon prices in the model as a result of limiting the availability of CCS. This contrasts with studies investigating global scenarios with low residual fossil emissions as a result of a broader portfolio of mitigation options (lifestyle changes, advanced technologies) relative to a standard mitigation scenario where these options are not available (Edelenbosch et al., 2024; Fuhrman et al., 2024).

Following previous EU scenario studies, our results confirm the importance of renewable power expansion, electrification, carbon-neutral fuels and CCS (Boitier et al., 2023; Rodrigues et al., 2022, 2022; van der Zwaan et al., 2025). While the role of e-fuels was mostly highlighted in scenarios with less pronounced demand-side electrification (Blanco et al., 2018; Evangelopoulou, 2019; Rodrigues et al., 2022; Schreyer et al., 2024), we show here that phasing out fossil fuels requires the simultaneous achievement of extensive electrification and drastic e-fuel scale-up. This enables the EU to reach climate neutrality with limited reliance on CO₂ storage (around 100 MtCO₂/yr), an important aspect given that the spatial distribution of biogenic CO₂ sources and prospective geological sinks may not allow for more than 200 MtCO₂/yr by 2050 (Rosa et al., 2021).

The comparison to literature underscores several limitations of our study. First, we do not specifically look at demand reduction measures that affect residual hydrocarbon use e.g. by circularity in the chemicals sector or lifestyle changes like reduced air travel (Grubler et al., 2018; Sharmina et al., 2020; Stegmann et al., 2022). Second, our model ignores technological options to switch away from hydrocarbon fuels in aviation and shipping. This assumption is nonetheless plausible considering the limited timeframe until 2050 and the nascent nature of these technologies

(Adu-Gyamfi and Good, 2022; Sharmina et al., 2020). Third, our scenarios assume that the EU land carbon sink from existing forests sustains its current absorption of 240 MtCO₂/yr, which is within the range of 100-400 MtCO₂/yr estimated by Pilli et al. (2022). Substantial uncertainty surrounds future forests (Pilli et al., 2022) and relates to management practices as well as climate change impacts (Verkerk et al., 2022). Moreover, land carbon removals are not permanent, which makes it problematic to fully account them as offsets for fossil CO₂ emissions (Carton et al., 2021). Fourth, our model aggregates the 27 EU countries into just 8 regions, which restricts our insight into the geographical aspects of the energy transition. While we do not expect fundamental changes in the observed dynamics, a higher spatial resolution will be needed to investigate national scenarios and cross-European energy grids (Breyer et al., 2022a; Hofmann et al., 2025; Neumann et al., 2023).

Our results have policy implications with respect to technology development and target setting in the EU. First, expansion of renewable power and electrification can lead a long way to the phase-out of fossil fuels, but technologies like CCS or e-fuel production will be needed to abate the last 10% of fossil fuels. Broad electrification is a necessary precondition for an extensive (90%) or nearly complete (>99%) phase-out of fossil fuels by 2050, with an electricity share in final energy close to 60% in all our scenarios. It emphasizes the importance of a policy strategy that clearly prioritizes electrification and reserves green hydrogen, e-fuels and CCS only for specific applications (Johnson et al., 2025; Kazlou et al., 2024; Schreyer et al., 2024; Ueckerdt et al., 2021; van der Zwaan et al., 2025).

In addition, substantial carbon capture of more than 200 MtCO₂/yr is required in all net-zero scenarios even as CO₂ storage is limited in *FosFree-NZ*. Investing in non-fossil carbon capture is therefore essential as it is needed for both e-fuel production and CDR via CO₂ storage. E-fuel production and CO₂ storage technologies should be developed in parallel to create robust pathways that secure against potential delays in up-scaling one of them. Given the ambitious scale-up of these technologies in our scenarios and current project delays (Kazlou et al., 2024; Odenweller and Ueckerdt, 2025), the window for reaching an extensive or nearly complete fossil phase-out by 2050 is rapidly closing unless increasingly large projects can be realized in the next years.

Second, the decision on an EU fossil phase-out target by 2050 hinges on a fundamental trade-off involving different types of risks and uncertainty. On the one hand, such a target would create an additional focal point for the EU energy transition and avoid deterring mitigation due to false expectations about the compatibility of fossil fuels with net-zero. It would provide clear signals to investors about the course of the energy transition, and help establish political credibility, a key ingredient to a functioning EU-ETS (Dolphin et al., 2023; Pahle et al., 2025; Sitarz et al., 2024). On the other hand, a full fossil phase-out comes with a considerable increase in marginal abatement costs. This is mainly driven by the need to rapidly scale-up e-fuel supply to 1000 TWh/yr in 2050, which is about as much as the total current EU fossil fuel consumption in aviation and shipping.

One viable middle course could entail defining partial and sector-specific targets for a fossil phase-out. Following the scenario ranges derived in our study, the EU could define a target of 90-100% reduction in fossil energy by 2050 relative to 2020 alongside the net-zero goal, and break it down to sectoral level by extending the existing framework of EU renewable energy targets (EU, 2023a). Nearly

From net-zero to zero-fossil in transforming the EU energy system
Revision 1, Nature Communications

complete phase-out targets could apply to the building sector, road transport and industrial process heat, while residual fossil use may be allowed in aviation, shipping and chemicals. The exact formulation and flexibility of such targets certainly calls for political negotiation, which could be a challenging process given their far-reaching implications and the divergent interests of stakeholders. Ultimately, this trade-off between stabilizing target expectations and maintaining market flexibility involves different layers of social, economic and political aspects that go beyond the scope of our energy system modeling perspective.

Several open questions remain to be addressed in future research on low-fossil energy systems. First, given the supply-side challenges to scale up non-fossil hydrocarbons, it is important to better understand how this pressure can be alleviated by demand-side measures like circular economy and lifestyle changes. Second, our study does not explore the dynamics around bioenergy provision and implications for land systems. Investigating the competition and trade-offs between e-fuel and bioenergy to supply non-fossil hydrocarbons would be of great value. Finally, the phase-out of fossil fuels was one of the key elements of the Conference of the Parties (COP) in Dubai in 2023. Spelling out global fossil phase-out scenarios that inform international climate negotiations will be particularly relevant in the run-up to the next global stocktake by 2028.

See end of this reviewer's comments for suggestion to restructure discussion.

I believe this is important and substantial work, but I highly encourage improving the discussion to meaningfully communicate the key messages, clear implications and recommendations, and strengthen novelty.

I have attached detailed comments with further suggestions. I look forward to receiving your responses and revised article.

Best wishes,

Lines Section Comment

1 20 Abstract You have some space to specify "REMIND integrated assessment model" which is useful for IAM-readers.

Included.

2 26-27 Abstract Reinforce that it is a complete fossil phase-out so it is not mistaken by general readers as the EUs current plans.

Included.

Also, consider the following the sentence structure that might be more impactful.

“Although posing additional transformation challenges, committing to a complete fossil phase-out would strengthen EU climate policy”

Thanks for the suggestion. The sentence now reads:

Our works shows the additional transformation challenges if the EU aims to strengthen its climate policy commitment with a full fossil phase-out target.

3 Introduction Double-check the referencing format of EU-related citations. There is an interchange between the EU, European Council, and EU Commission. These may be correct, but these documents can often be incorrectly referenced; the EU might need to be spelled out to “European Union”.

We see your point. We checked and updated all EU-related citations. We used EU for legislated directives and regulations. For statements by the European Council or plans by the EU Commission we kept the references as they are. If desired by the journal, we can spell out EU as European Union in references. Please find the updated references below:

EU, 2024. Regulation (EU) 2024/1735 of the European Parliament and of the Council of 13 June 2024 on establishing a framework of measures for strengthening Europe’s net-zero technology manufacturing ecosystem and amending Regulation (EU) 2018/1724Text with EEA relevance.

EU, 2023a. Directive (EU) 2023/2413 of the European Parliament and of the Council of 18 October 2023 amending Directive (EU) 2018/2001, Regulation (EU) 2018/1999 and Directive 98/70/EC as regards the promotion of energy from renewable sources, and repealing Council Directive (EU) 2015/652.

EU, 2023b. Directive of the European Parliament and of The Council amending Directive 2003/87/EC establishing a system for greenhouse gas emission allowance trading within the Union and Decision (EU) 2015/1814 concerning the establishment and operation of a market stability reserve for the Union greenhouse gas emission trading system.

EU, 2023c. Commission Delegated Regulation (EU) of 10.2.2023 supplementing Directive (EU) 2018/2001 of the European Parliament and of the Council by establishing a Union methodology setting out detailed rules for the production of renewable liquid and gaseous transport fuels of non-biological origin.

EU, 2023d. Regulation (EU) 2023/851 of the European Parliament and of The Council of 19 April 2023 amending Regulation (EU) 2019/631 as regards strengthening the CO2 emission performance standards for new passenger cars and new light commercial vehicles in line with the Union’s increased climate ambition.

EU, 2021. European Climate Law. Regulation (EU) 2021/1119 of the European Parliament and of the Council of 30 June 2021 Establishing the Framework for Achieving Climate Neutrality and Amending Regulations (EC) No 401/2009 and (EU) 2018/1999.

EU Commission, 2022. European Commission - Press release REPowerEU: A plan to rapidly reduce dependence on Russian fossil fuels and fast forward the green transition.

EU Commission, 2018. In-depth Analysis in support of the Commission - A Clean Planet for all - A European long-term strategic vision for a prosperous, modern, competitive and climate neutral economy (No. COMMUNICATION COM(2018) 773). European Commission, Brussels.

European Council, 2023. Preparations for the 28th Conference of the Parties (COP28) of the United Nations Framework Convention on Climate Change (UNFCCC).

4 38 Introduction Perhaps it is more clear to write “However, this ambition has not translated into concrete EU targets for completely reducing or phasing-out fossil fuel consumption.”

Since the EU Green Deal and related material do suggest fossil phase-out by adoption of several renewable technologies etc. it is perhaps in the paper's interest to focus on the complete phase-out aspect.

Thanks. We changed it to “for reducing or completely phasing out fossil fuel consumption”. As argued in the discussion section, it would also be a step to define incremental fossil reduction targets even if they are not a 100%.

5 44 Introduction Check if there is an accidental additional space between “available remains”

Done.

6 51 Introduction This is a great sentence, but it is also a generalisation. Are you able to pull a citation or case-study example to strengthen this? If not, add some caution to the sentence such as:

“This poses the risk of creating wrong expectations as stakeholders in the energy sector may scale down mitigation ambition in anticipation of sufficient future CCS or CDR to abate their fossil emissions.”

Thanks, we went with your suggestion.

7 87-99, 647 Results, Methods Please can you justify why, in the LeastCost-NZ scenario, the EU land carbon sink is assumed to jump to 370 MtCO₂/yr in 2050 suddenly, but in the FosFree-NZ scenario, the land carbon sink remains at 240 MtCO₂/yr in 2050?

This seems like an assumption that should be consistent; it implies that the LeastCost-NZ scenario will have substantial changes in biomass, soil carbon, forests, etc., immediately in 2050. Moreover, both scenarios have the same biomass consumptions in Figure 1

A substantial 130 MtCO₂/yr (+50%) sudden increase likely impacts the solution space for REMIND. Please provide substantial justification for this assumption with evidence or sensitivity testing of the value at 240 MtCO₂/yr etc.

“First, we model a least-cost NZ scenario (LeastCost-NZ) where CO₂ storage injection is free to scale up to 2 Gt/yr in the EU at low injection costs. Furthermore, we optimistically assume that the EU land carbon sink can increase from the current level of 240 MtCO₂/yr to 370 MtCO₂/yr in 2050, exceeding the 2030 goal of 310 MtCO₂/yr (European Commission, 2021). This is our benchmark net-zero scenario without a fossil phase-out. Second, we model a fossil-free NZ scenario (FosFree-NZ) where CO₂ storage injection is limited to 110 MtCO₂/yr, which is the lowest setting for the model to still reach the net-zero target, with the land carbon sink remaining at 240 MtCO₂/yr.”

“We make an optimistic assumption of a carbon removal of 370 MtCO₂/yr by 2050 within the range of Pilli et al. (2022), which is slightly larger than what was used for the EU Commission (2018).”

Thanks for thinking about this more deeply. You are right that this is an important assumption and needs justification. The initial idea was to show a scenario with more CDR contribution from existing forests that also aligns with the LULUCF target of the European Union and substantially enhances the land sink relative to today. This would in turn allow for more fossil fuels at net-zero. But as we neither model the land sink of existing forests, nor have costs attributed to such an increase, we decided to remove the scenario with more CDR from existing forests from the standard scenarios for conceptual clarity. Instead, we add this scenario with optimistic land sink assumptions to the sensitivity analysis.

This means, that the updated Least-cost NZ scenario only features a higher limit of CCS injection (2Gt CO₂/yr) relative to the FossFree-NZ scenario (with 110 MtCO₂/yr CCS limit), while all other assumptions remain the same (land sink at 240 MtCO₂/yr).

You are right that enhancing the land sink could possibly compete with the potential for bioenergy production. Regarding bioenergy we go with the sustainable low potential as estimated by Ruiz et al. (2019) (7.5 EJ/yr) for the standard scenarios as well as the sensitivity scenarios with the higher land sink.

We justify the precautionary CDR assumption from existing forests of 240 MtCO₂/yr with the following addition to the discussion section:

Third, our scenarios assume that the EU land carbon sink from existing forests sustains its current absorption of 240 MtCO₂/yr, which is within the range of 100-400 MtCO₂/yr estimated by Pilli et al. (2022). Substantial uncertainty surrounds future forests (Pilli et al., 2022) and relates to management practices as well as climate change impacts (Verkerk et al., 2022). Moreover, land carbon removals are not permanent, which makes it problematic to fully account them as offsets for fossil CO₂ emissions (Carton et al., 2021).

8 101-110 Results, Figure 1 As discussed in the text, the FosFree-NZ and LeastCost-NZ scenarios have considerably more optimistic fossil reductions compared to the 1.5-2.0C pathways. The key drivers of this are discussed in the following paragraph and Figure 1. However, in the discussion section, please discuss the model factors and assumptions contributing to these differences compared to other models that feed the IPCC scenarios. This can be at a general level, but the study results are based on the assumptions and limitations of REMIND, and it's an important discussion point. One example is that REMIND is based on perfect foresight and long-term decisions, meaning it is more likely to select the most optimum and optimistic technology solutions compared to other models (e.g. rapid solar, electrification). In contrast, IMAGE has myopic foresight is focused on short-term decisions and is more likely to lead to more pessimistic solutions. Below, are a few references that could help with this discussion (Please note, I have no relationship to these articles)

Dekker, M.M., Daioglou, V., Pietzcker, R. et al. Identifying energy model fingerprints in mitigation scenarios. *NatEnergy* 8, 1395–1404 (2023). <https://doi.org/10.1038/s41560-023-01399-1>

Mathijs Harmsen et al 2021 *Environ. Res. Lett.* 16 054046 Integrated assessment model diagnostics: key indicators and model evolution 10.1088/1748-9326/abf964

I Keppo et al 2021 *Environ. Res. Lett.* 16 053006 Exploring the possibility space: taking stock of the diverse capabilities and gaps in integrated assessment models 10.1088/1748-9326/abe5d8

Thanks for this point. We agree that this requires further discussion. We address this in the revised discussion section:

Our IAM study is the first to explore a complete phase-out of fossil fuels by 2050 in the EU. Scenarios from the AR6 database typically show substantially higher fossil fuel use by mid-century and beyond. First, diverging scenario designs can explain this difference: Unlike AR6 scenarios compatible with 1.5-2°C global warming, our study imposes net-zero GHG emissions in the EU and exogenous limits to CCS. Second, each model has its own characteristics, and REMIND tends to be particularly reactive to climate policies in terms of emissions abatement (Dekker et al., 2023), featuring high renewable power and electrification shares and limited use of fossil fuels and CCS (Luderer et al., 2022; Soergel et al., 2021). Methodologically, it is important to understand that the full fossil-phase-out is triggered by an endogenous response to increasing carbon prices in the model as a result of limiting the availability of CCS. This contrasts with studies investigating global scenarios with low residual fossil emissions as a result of a broader portfolio of mitigation options (lifestyle changes, advanced technologies) relative to a standard mitigation scenario where these options are not available (Edelenbosch et al., 2024; Fuhrman et al., 2024).

Following previous EU scenario studies, our results confirm the importance of renewable power expansion, electrification, carbon-neutral fuels and CCS (Boitier et al., 2023; Rodrigues et al., 2022, 2022; van der Zwaan et al., 2025). While the role of e-fuels was mostly highlighted in scenarios with less pronounced demand-side electrification (Blanco et al., 2018; Evangelopoulou, 2019; Rodrigues et al., 2022; Schreyer et al., 2024), we show here that phasing out fossil fuels requires the simultaneous achievement of extensive electrification and drastic e-fuel scale-up. This enables the EU to reach climate neutrality with limited reliance on CO₂ storage (around 100 MtCO₂/yr), an important aspect given that the spatial distribution of biogenic CO₂ sources and prospective geological sinks may not allow for more than 200 MtCO₂/yr by 2050 (Rosa et al., 2021).

The comparison to literature underscores several limitations of our study. First, we do not specifically look at demand reduction measures that affect residual hydrocarbon use e.g. by circularity in the chemicals sector or lifestyle changes like reduced air travel (Grubler et al., 2018; Sharmina et al., 2020; Stegmann et al., 2022). Second, our model ignores technological options to switch away from hydrocarbon fuels in aviation and shipping. This assumption is nonetheless plausible considering the limited timeframe until 2050 and the nascent nature of these technologies (Adu-Gyamfi and Good, 2022; Sharmina et al., 2020). Third, our scenarios assume that the EU land carbon sink from existing forests sustains its current absorption of 240 MtCO₂/yr, which is within the range of 100-400 MtCO₂/yr estimated by Pilli et al. (2022). Substantial uncertainty surrounds future forests (Pilli et al., 2022) and relates to management practices as well as climate change impacts (Verkerk et al., 2022). Moreover, land carbon removals are not permanent, which makes it problematic to fully account them as offsets for fossil CO₂ emissions (Carton et al., 2021). Fourth, our model aggregates the 27 EU countries into just 8 regions, which restricts our insight into the geographical aspects of the energy transition. While we do not expect fundamental changes in the observed dynamics, a higher spatial resolution will be needed to investigate national scenarios and cross-European energy grids (Breyer et al., 2022a; Hofmann et al., 2025; Neumann et al., 2023).

9 Figure 2 This is a nice Sankey, but it is slightly hard to follow and compare between the three. I would suggest to ensure the ordering on the left and the right stays the same to help readers compare. For example, ensure the fossil source (leftmost) stays the top, and apply to the others. Moreover, in the EU27 LeastCost-NZ 2050 (Middle), the final part of the secondary energy value for hydrocarbons is hidden (248_), please ensure this is fully visible.

We thank the reviewer for the visual suggestions. We worked on the Sankey diagrams to make them easier to read and to compare: the ordering is consistent across scenarios, the font is bigger and more contrasted, and minor links have been removed for clarity.

10 Figure 3 This is an insightful figure that clearly illustrates the scenario differences, particularly from 2035 which seems to be a critical point. It could benefit from an additional chart related to CCS/CDR deployment and could perhaps be included on a secondary axis. It would help understand CCS/CDR deployment over time.

We see the benefit of this. However, as Figure 3 already has a lot of panels we decided to add this plot to the Supplementary Information section. We included it in the main text in the following way:

[...] This full de-fossilization of residual hydrocarbons in the last 10-15 years is the main difference between the *LeastCost-NZ* and the *FosFree-NZ* in the energy sector, which show similar levels of total hydrocarbon demand as the *LeastCost-NZ* already deploys most of the available electrification options until 2050 (Figure 3f, Ext Data Fig 4). To reach net-zero in the *LeastCost-NZ* scenario with residual fossil energy, there is more CDR via increased usage of bioenergy with carbon capture and storage (Supplementary Figs. 1 and 2).

From net-zero to zero-fossil in transforming the EU energy system
Revision 1, Nature Communications

SI Figure 1: Total carbon dioxide removal in the LeastCost-NZ and the FosFree-NZ scenarios (thick lines) as well as sensitivity scenarios (thin lines, green funnel) in the EU up to 2050. Carbon dioxide removal in this analysis includes the land carbon sink, bioenergy with carbon capture and storage as well as direct air capture with carbon capture and storage.

11 205-206 Results Please could you check and clarify that this sentence makes sense. I'm not sure what you mean by e-fuels being deployed once CDR becomes limited?

"As e-fuels are an expensive mitigation option, they are only deployed at scale once CDR options like BECCS and DACCS are limited, which aligns with previous research"

We rephrased this to:

As e-fuels are an expensive mitigation option, they are only economical at scale once CDR options like BECCS and DACCS are not available for further emissions abatement, which aligns with previous research (Lehtveer et al., 2019; Mignone et al., 2024; Oshiro & Fujimori, 2022).

12 235-270 Results, Figure 5 Focussing on the leftmost chart, there is a substantial MAC jump from ~250 to ~500 EUR/tCO₂, for limiting FFCC from 400 to 300 MtCO₂/yr. What is the reason for the substantial initial jump in MAC in the standard model? In general, I think this section could be strengthened with a few sentences to describe the MAC trends (initial spike, plateau, spike again – from higher to lower FFCC) rather than just from the end-to-end.

Thank you, that is very well observed. This and other points led us to reconsider the scenario design. The reason for this MAC jump from 300 to 400 MtCO₂/yr was the increase of the land sink between those two scenarios according to our previous scenario design. The marginal abatement cost not only depend on FFCC but also on the type of CDR option used to reach net-zero. While increasing bioenergy with carbon capture and storage (BECCS) comes at a high cost, the land sink is an exogenous assumption which comes at no abatement cost to the model. In the LeastCost-NZ setting which allows for high CCS deployment, increasing the land sink does not only allow for more residual fossils, but also enables less BECCS to reach net-zero as more land CDR is available. This disproportionately decreases the marginal abatement cost. Together with the aforementioned uncertainty about enhancing the EU land carbon sink, this led us to change our scenario design to only feature variations of the CCS injection rate as this avoids artefacts in the shape of the MAC curve. As you can see in the *HighLandSink* sensitivity scenarios in panel B) below, assuming the large land sink across all scenarios leads to an overall lower marginal abatement cost curve. We added a note on the convex shape of the MAC curves to the main text:

[...] Higher CO₂ capture rates (*HighCaptureRate*) and variations on the cost of direct air capture (*CheapDAC*) hardly affect MAC in *FossFree-NZ*, and only slightly reduce them in *LeastCost-NZ* with high CCS availability (see Ext Data Fig 1). Generally, the MAC curves tend to follow a convex shape with stronger cost increases at low FFCC. Overall, the sensitivity analyses indicate that CO₂ prices required for a full fossil phase-out until mid-century strongly depend on the availability of low-cost carbon-neutral fuels.

Figure 1: Marginal abatement cost and aggregate consumption losses of EU net-zero scenarios with varying levels of residual fossil fuels. A) EU marginal abatement cost on the y-axis against fossil fuel carbon consumption (carbon contained

From net-zero to zero-fossil in transforming the EU energy system
Revision 1, Nature Communications

in total demand of fossil fuels) in 2050 on the x-axis across NZ standard scenarios (black line) and corresponding fuel-switching CO₂ prices for replacing fossil liquid fuels by e-fuels based shadow prices from the model (dashed red line). B) EU marginal abatement as in A) across all NZ scenarios (sensitivity scenario in colored lines). C) EU aggregate 2025-2050 consumption losses discounted at 3% per year of all NZ scenarios (sensitivity scenario in colored lines) relative to the Weak Policy scenario (see Methods section M2) on the y-axis against fossil fuel carbon in 2050 on the x-axis. See section M5 Weak Policy Scenario and Consumption Losses **Error! Reference source not found.** for details. See M2. Scenario Assumptions and Ext Data Table 1 for description and categorization of scenarios.

13 Discussion The discussion could be significantly strengthened, as it mainly summarises the results. The study results are important, can have a significant impact, and could land much more effectively with an improved discussion. Here are some suggestions to consider:

- A quick summary is fine, but I think it could be shorted.
- 338-347 discusses some assumptions and limitations nicely, but I would recommend including discussion on the uncertainty and limitations of using only REMIND (see comment 7+8), and how outcomes can differ from different IAMs. Moreover, it may be useful to discuss the assumptions on land carbon sink, and what other parameters could have been sensitivity testing, and uncertainty of regional disaggregation since you mention REMIND contains 8 EU regions in the methodology.
- Can you discuss and draw some key comparisons compared to other literature mentioned in the introduction? A multi-model analysis of the EU's path to net zero, Boitier, Baptiste et al. *Joule*, Volume 7, Issue 12, 2760 – 2782 <https://doi.org/10.1016/j.joule.2023.11.002>.

I was also surprised, this paper was not included, which can provide some relevant discussion points too. Ueckerdt, F., Bauer, C., Dirnaichner, A. et al. Potential and risks of hydrogen-based e-fuels in climate change mitigation. *Nat. Clim. Chang.* 11, 384–393 (2021). <https://doi.org/10.1038/s41558-021-01032-7>.

- The final paragraph discusses implications but also shies away from making clearer and meaningful recommendations that policy-makers or practitioners can take away. The core of the paper is attempting to show pathways to zero fossil, and it would be beneficial at the end to have a few bullets or numbered items on key recommendations/suggestions e.g. strengthening support/investment/import for carbon-neutral fuels development for hard-to-decarbonise sectors.

We thank the reviewer for this helpful and constructive feedback. We improved the discussion with respect to literature comparisons and policy implications. It now reads as follows:

This study investigates the transformation dynamics and challenges of a phase-out of fossil fuels in the EU along with the climate neutrality goal by 2050. In summary, we find that in a least-cost net-zero scenario (*LeastCost-NZ*) without exogenous limits on carbon capture and storage (CCS), fossil fuel consumption already decreases by 90% in 2050 relative to 2020 at MAC of 460 EUR/tCO₂. Residual fossil fuels are mainly oil-based liquids and some natural gas. They can be phased-out with additional transformation efforts on the energy supply side by massively scaling up carbon-neutral e-fuels, which increases the MAC to 630 EUR/tCO₂. In such a full fossil phase-out scenario (*FosFree-NZ*), MAC are particularly sensitive to assumptions about bioenergy availability, e-fuel imports and technology scalability (500-1000 EUR/tCO₂ range). In comparison, Victoria et al. (2022) find MAC in 2050 of around 300 EUR/tCO₂ in scenarios with ambitious EU climate targets and extremely fast technology up-scaling. In REMIND, we model more realistic up-scaling as technology costs increase non-linearly with increasing growth rates. In ECEMF scenarios (see Figure 1), EU carbon prices also tend to be

around 600 EUR/tCO₂ in 2050, yet in scenarios with higher residual fossil fuel use compared to our *FosFree-NZ* scenario. Despite steeply increasing MAC to replace the last 10% of fossil fuels, aggregate economic costs do not increase as much in our scenarios since high MAC are only incurred in specific sectors with a small share in the total economy. This suggests that transformational challenges may be managed more easily if concerns about the cost distribution are addressed.

Our IAM study goes beyond existing scenario literature by exploring a complete phase-out of fossil fuels by 2050 in the EU. Scenarios from the AR6 database typically show substantially higher fossil fuel use by mid-century and beyond. First, diverging scenario designs can explain this difference: Unlike AR6 scenarios compatible with 1.5-2°C global warming, our study imposes net-zero GHG emissions in the EU and exogenous limits to CCS. Second, each model has its own characteristics, and REMIND tends to be particularly reactive to climate policies in terms of emissions abatement (Dekker et al., 2023), featuring high renewable power and electrification shares and limited use of fossil fuels and CCS (Luderer et al., 2022; Soergel et al., 2021). Methodologically, it is important to understand that the full fossil-phase-out is triggered by an endogenous response to increasing carbon prices in the model as a result of limiting the availability of CCS. This contrasts with studies investigating global scenarios with low residual fossil emissions as a result of a broader portfolio of mitigation options (lifestyle changes, advanced technologies) relative to a standard mitigation scenario where these options are not available (Edelenbosch et al., 2024; Fuhrman et al., 2024).

Following previous EU scenario studies, our results confirm the importance of renewable power expansion, electrification, carbon-neutral fuels and CCS (Boitier et al., 2023; Rodrigues et al., 2022, 2022; van der Zwaan et al., 2025). While the role of e-fuels was mostly highlighted in scenarios with less pronounced demand-side electrification (Blanco et al., 2018; Evangelopoulou, 2019; Rodrigues et al., 2022; Schreyer et al., 2024), we show here that phasing out fossil fuels requires the simultaneous achievement of extensive electrification and drastic e-fuel scale-up. This enables the EU to reach climate neutrality with limited reliance on CO₂ storage (around 100 MtCO₂/yr), an important aspect given that the spatial distribution of biogenic CO₂ sources and prospective geological sinks may not allow for more than 200 MtCO₂/yr by 2050 (Rosa et al., 2021).

The comparison to literature underscores several limitations of our study. First, we do not specifically look at demand reduction measures that affect residual hydrocarbon use e.g. by circularity in the chemicals sector or lifestyle changes like reduced air travel (Grubler et al., 2018; Sharmina et al., 2020; Stegmann et al., 2022). Second, our model ignores technological options to switch away from hydrocarbon fuels in aviation and shipping. This assumption is nonetheless plausible considering the limited timeframe until 2050 and the nascent nature of these technologies (Adu-Gyamfi and Good, 2022; Sharmina et al., 2020). Third, our scenarios assume that the EU land carbon sink from existing forests sustains its current absorption of 240 MtCO₂/yr, which is within the range of 100-400 MtCO₂/yr estimated by Pilli et al. (2022). Substantial uncertainty surrounds future forests (Pilli et al., 2022) and relates to management practices as well as climate change impacts (Verkerk et al., 2022). Moreover, land carbon removals are not permanent, which makes it problematic to fully account them as offsets for fossil CO₂ emissions (Carton et al., 2021). Fourth, our model aggregates the 27 EU countries into just 8 regions, which restricts our insight into the geographical aspects of the energy transition. While we do not expect fundamental changes in the

observed dynamics, a higher spatial resolution will be needed to investigate national scenarios and cross-European energy grids (Breyer et al., 2022a; Hofmann et al., 2025; Neumann et al., 2023).

Our results have policy implications with respect to technology development and target setting in the EU. First, expansion of renewable power and electrification can lead a long way to the phase-out of fossil fuels, but technologies like CCS or e-fuel production will be needed to abate the last 10% of fossil fuels. Broad electrification is a necessary precondition for an extensive (90%) or nearly complete (>99%) phase-out of fossil fuels by 2050, with an electricity share in final energy close to 60% in all our scenarios. It emphasizes the importance of a policy strategy that clearly prioritizes electrification and reserves green hydrogen, e-fuels and CCS only for specific applications (Johnson et al., 2025; Kazlou et al., 2024; Schreyer et al., 2024; Ueckerdt et al., 2021; van der Zwaan et al., 2025).

In addition, substantial carbon capture of more than 200 MtCO₂/yr is required in all net-zero scenarios even as CO₂ storage is limited in *FosFree-NZ*. Investing in non-fossil carbon capture is therefore essential as it is needed for both e-fuel production and CDR via CO₂ storage. E-fuel production and CO₂ storage technologies should be developed in parallel to create robust pathways that secure against potential delays in up-scaling one of them. Given the ambitious scale-up of these technologies in our scenarios and current project delays (Kazlou et al., 2024; Odenweller and Ueckerdt, 2025), the window for reaching an extensive or nearly complete fossil phase-out by 2050 is rapidly closing unless increasingly large projects can be realized in the next years.

Second, the decision on an EU fossil phase-out target by 2050 hinges on a fundamental trade-off involving different types of risks and uncertainty. On the one hand, such a target would create an additional focal point for the EU energy transition and avoid deterring mitigation due to false expectations about the compatibility of fossil fuels with net-zero. It would provide clear signals to investors about the course of the energy transition, and help establish political credibility, a key ingredient to a functioning EU-ETS (Dolphin et al., 2023; Pahle et al., 2025; Sitarz et al., 2024). On the other hand, a full fossil phase-out comes with a considerable increase in marginal abatement costs. This is mainly driven by the need to rapidly scale-up e-fuel supply to 1000 TWh/yr in 2050, which is about as much as the total current EU fossil fuel consumption in aviation and shipping.

One viable middle course could entail defining partial and sector-specific targets for a fossil phase-out. Following the scenario ranges derived in our study, the EU could define a target of 90-100% reduction in fossil energy by 2050 relative to 2020 alongside the net-zero goal, and break it down to sectoral level by extending the existing framework of EU renewable energy targets (EU, 2023a). Nearly complete phase-out targets could apply to the building sector, road transport and industrial process heat, while residual fossil use may be allowed in aviation, shipping and chemicals. The exact formulation and flexibility of such targets certainly calls for political negotiation, which could be a challenging process given their far-reaching implications and the divergent interests of stakeholders. Ultimately, this trade-off between stabilizing target expectations and maintaining market flexibility involves different layers of social, economic and political aspects that go beyond the scope of our energy system modeling perspective.

Several open questions remain to be addressed in future research on low-fossil energy systems. First, given the supply-side challenges to scale up non-fossil hydrocarbons, it is important to better understand how this pressure can be alleviated by demand-side measures like circular economy and lifestyle changes. Second, our study does not explore the dynamics around bioenergy provision and implications for land systems. Investigating the competition and trade-offs between e-fuel and bioenergy to supply non-fossil hydrocarbons would be of great value. Finally, the phase-out of fossil fuels was one of the key elements of the Conference of the Parties (COP) in Dubai in 2023. Spelling out global fossil phase-out scenarios that inform international climate negotiations will be particularly relevant in the run-up to the next global stocktake by 2028.

14 552 Methods I think this is the more relevant definition “REMIND is an energy-economy-environment model” since REMIND and other IAMs contain environment modules etc.

We deliberately used the term energy-economy model as the REMIND version we use does not include an assessment of climate impacts. Moreover, it only features a simplified representation of the land system. To study the co-evolution of energy and land systems, REMIND can be run in a coupled mode with the model MAGPIE as done in other publications (Klein et al., 2014; Luderer et al., 2022; Soergel et al., 2021).

15 729 Methods I think you mean 2030?

No, 2030 is meant. This is the Weak Policy scenario based on current trends that does not reach the EU climate targets. We clarified the sentence:

This *Weak Policy* scenario does not reach the EU climate targets. It reduces total GHG emissions to only about 2200 MtCO₂/yr by 2050, which is the level of 55% reductions relative to 1990 that is supposed to be reached already in 2030 according to the EU climate targets.

Reviewer #2

This is an excellent and very timely paper assessing fossil fuel phase-out in Europe. As far as I can tell the methodology is solid and well-explained. The results are plausible. The only suggestion for revisions I have is to at least briefly discuss the policy implications of this work in more detail and be more specific about the "so what?" question.

Thank you for this positive feedback. Based on your comment and those by other reviewers, we worked over the discussion section and put more emphases on policy implications. The section on policy implications reads as follows:

Our results have policy implications with respect to technology development and target setting in the EU. First, expansion of renewable power and electrification can lead a long way to the phase-out of fossil fuels, but technologies like CCS or e-fuel production will be needed to abate the last 10% of fossil fuels. Broad electrification is a necessary precondition for an extensive (90%) or nearly complete (>99%) phase-out of fossil fuels by 2050, with an electricity share in final energy close to 60% in all our scenarios. It emphasizes the importance of a policy strategy that clearly prioritizes electrification and reserves green hydrogen, e-fuels and CCS only for specific applications (Johnson et al., 2025; Kazlou et al., 2024; Schreyer et al., 2024; Ueckerdt et al., 2021; van der Zwaan et al., 2025).

In addition, substantial carbon capture of more than 200 MtCO₂/yr is required in all net-zero scenarios even as CO₂ storage is limited in *FosFree-NZ*. Investing in non-fossil carbon capture is therefore essential as it is needed for both e-fuel production and CDR via CO₂ storage. E-fuel production and CO₂ storage technologies should be developed in parallel to create robust pathways that secure against potential delays in up-scaling one of them. Given the ambitious scale-up of these technologies in our scenarios and current project delays (Kazlou et al., 2024; Odenweller and Ueckerdt, 2025), the window for reaching an extensive or nearly complete fossil phase-out by 2050 is rapidly closing unless increasingly large projects can be realized in the next years.

Second, the decision on an EU fossil phase-out target by 2050 hinges on a fundamental trade-off involving different types of risks and uncertainty. On the one hand, such a target would create an additional focal point for the EU energy transition and avoid deterring mitigation due to false expectations about the compatibility of fossil fuels with net-zero. It would provide clear signals to investors about the course of the energy transition, and help establish political credibility, a key ingredient to a functioning EU-ETS (Dolphin et al., 2023; Pahle et al., 2025; Sitarz et al., 2024). On the other hand, a full fossil phase-out comes with a considerable increase in marginal abatement costs. This is mainly driven by the need to rapidly scale-up e-fuel supply to 1000 TWh/yr in 2050, which is about as much as the total current EU fossil fuel consumption in aviation and shipping.

One viable middle course could entail defining partial and sector-specific targets for a fossil phase-out. Following the scenario ranges derived in our study, the EU could define a target of 90-100% reduction in fossil energy by 2050 relative to 2020 alongside the net-zero goal, and break it down to sectoral level by extending the existing framework of EU renewable energy targets (EU, 2023a). Nearly complete phase-out targets could apply to the building sector, road transport and industrial process heat, while residual fossil use may be allowed in aviation, shipping and chemicals. The exact

formulation and flexibility of such targets certainly calls for political negotiation, which could be a challenging process given their far-reaching implications and the divergent interests of stakeholders. Ultimately, this trade-off between stabilizing target expectations and maintaining market flexibility involves different layers of social, economic and political aspects that go beyond the scope of our energy system modeling perspective.

Reviewer #3

The manuscript deals with an interesting topic that is consistent with the purpose of the journal. The manuscript has a suitable content for a scientific paper. The structure of the paper is good. The Data and Methodology section is also very well written in sufficient detail. The Results are also presented in a straightforward and clear way. The discussion section is good.

The manuscript has potential to be published but specific issues need to be addressed before publication.

1. In the abstract, authors must present the method used and the period of the study.

Thanks for the feedback. We modified the third sentence of the abstract to clarify the method (integrated assessment modeling) and the time period investigated (up to 2050).

Using the integrated assessment model REMIND, we quantify the additional effort needed to achieve a nearly complete EU-wide phase-out of fossil fuels by 2050 compared to a least-cost net-zero scenario.

2. The authors must present in the final part of the study, the limits of research and future directions of research.

We welcome the reviewer's point. We adapted our discussion section in several ways. It now features a more comprehensive part on limitations as well as some thoughts on future research opportunities:

The comparison to literature underscores several limitations of our study. First, we do not specifically look at demand reduction measures that affect residual hydrocarbon use e.g. by circularity in the chemicals sector or lifestyle changes like reduced air travel (Grubler et al., 2018; Sharmina et al., 2020; Stegmann et al., 2022). Second, our model ignores technological options to switch away from hydrocarbon fuels in aviation and shipping. This assumption is nonetheless plausible considering the limited timeframe until 2050 and the nascent nature of these technologies (Adu-Gyamfi and Good, 2022; Sharmina et al., 2020). Third, our scenarios assume that the EU land carbon sink from existing forests sustains its current absorption of 240 MtCO₂/yr, which is within the range of 100-400 MtCO₂/yr estimated by Pilli et al. (2022). Substantial uncertainty surrounds future forests (Pilli et al., 2022) and relates to management practices as well as climate change impacts (Verkerk et al., 2022). Moreover, land carbon removals are not permanent, which makes it

From net-zero to zero-fossil in transforming the EU energy system
Revision 1, Nature Communications

problematic to fully account them as offsets for fossil CO₂ emissions (Carton et al., 2021). Fourth, our model aggregates the 27 EU countries into just 8 regions, which restricts our insight into the geographical aspects of the energy transition. While we do not expect fundamental changes in the observed dynamics, a higher spatial resolution will be needed to investigate national scenarios and cross-European energy grids (Breyer et al., 2022a; Hofmann et al., 2025; Neumann et al., 2023).

[...]

Several open questions remain to be addressed in future research on low-fossil energy systems. First, given the supply-side challenges to scale up non-fossil hydrocarbons, it is important to better understand how this pressure can be alleviated by demand-side measures like circular economy and lifestyle changes. Second, our study does not explore the dynamics around bioenergy provision and implications for land systems. Investigating the competition and trade-offs between e-fuel and bioenergy to supply non-fossil hydrocarbons would be of great value. Finally, the phase-out of fossil fuels was one of the key elements of the Conference of the Parties (COP) in Dubai in 2023. Spelling out global fossil phase-out scenarios that inform international climate negotiations will be particularly relevant in the run-up to the next global stocktake by 2028.

Reviewer #4

This paper uses the REWIND IAM to model scenarios to achieve the EU 2050 economy-wide decarbonization goal. The authors construct a least-cost scenario, in which some fossil fuel is used, that serves as a baseline against which other scenarios are judged. This least-cost path achieves net-zero in part by employing CCS, CDR, and some carbon-neutral fuels and e-fuels. The objective of the paper is to determine a scenario that achieves net-zero by phasing out these remaining 10-15% of fossil fuels. The resulting fossil-free scenario has an abatement cost that is twice as much as that for the least-cost scenario and is differentiated from the other scenario by two important characteristics: greater e-fuels utilization and less CCS deployment. The authors then look at the projected growth rate for these technologies in reference to the historical growth rates of energy sources. This comparison demonstrates that full deployment of each technology by 2050 may be difficult to achieve. The authors conclude that while a goal of no fossil fuel may be a useful focal point for decarbonization, a better course may be to allow fossil fuel quotas for specific sectors to add planning certainty and relieve pressure on the power sector.

This paper is a thoughtful and well written piece on a pertinent topic that is also discussed in many papers from different perspectives. The comparison of the model outputs illustrates the differences in energy and technology reliance and is something that can be leaned on more in the discussion. While the premise of the paper concerns policy decisions, no concrete policy recommendations are made. There seem to be opportunities for this regarding aspects such as DACS, importing e-fuels, and biomass. If these cannot be discussed here, I look forward to reading the next paper that does so. This reviewer finds that the article can be published in Nature Communications after some minor revisions.

Questions and recommendations for the authors

1. In the abstract, the authors emphasize that one strength of the complete phase-out of fossil fuels is that it shows EU commitment to the climate target. The weakness is that it also requires transformation challenges. In the conclusion, the authors end with e-fuels can take pressure off CCS deployment, which is suggesting that some combination of both should be done to avoid the difficulties of large-scale e-fuel and CCS deployment and make it more likely that the target can be achieved. Why does the abstract not mention the least-cost CCS option and the strength of both approaches? Word count is not a sufficient answer.

Thanks for this input. We suppose you refer to this statement at the end of the abstract:

[...] Although a full fossil phase-out target could strengthen EU climate policy commitment, it also poses additional transformation challenges.

The reason why we chose to highlight this trade-off is that the advantage of the LeastCost-NZ scenario with overall lower costs is already in its name and is what is typically emphasized by IAM scenarios (Achakulwisut et al., 2023; Kazlou et al., 2024). The novelty of the paper is centered around the full fossil phase-out scenario whose opportunities and challenges need to be explained to the reader.

Based on other suggestions, this sentence has been slightly reformulated:

Our works shows the additional transformation challenges if the EU aims to strengthen its climate policy commitment with a full fossil phase-out target.

2. Fig 1 should have same y-axes when possible.

Thanks for the input. But, we chose to leave the y-axis scaling as it is. Making the panels B) to K) have the same y-axes scale for percentage numbers would make the differences between scenarios hard to read. The focus of this plot is on comparing scenarios in a panel and not so much on comparing energy shares across panels, which represent different indicators.

3. In Figure 1, do the authors' scenarios directly compare to any of the AR6 scenarios, other than fuel use? What are the similarities? Is there a lesson from the common characteristics like more electrification is better, more hydrogen or a contrast in CCS and CDR?

We read the question as to what extent our scenarios differ from the AR6 scenarios. We improved the paragraph describing Figure 1 with regards to this aspect:

Looking at a broad set of indicators in 2050, the deep fossil fuel reduction in our EU net-zero scenarios is enabled by an accelerated energy transition until mid-century, going beyond most 1.5°C scenarios in the AR6 database (Figure 1b-k). Direct electrification of energy end-use in buildings, road transport, and industrial process heat is more pronounced as the electricity share in final energy of our scenarios increases to close to 60% in 2050 (Figure 1g) compared to a median of 47% in the AR6 scenarios. Our scenarios also show higher variable renewable energy (VRE) shares of up to 83% in the electricity mix (Figure 1e) that constitute the backbone of an emissions-free power system by mid-century. As a consequence of increased direct electrification, the share of hydrocarbon energy carriers in final energy decreases substantially (from 75% in 2020 to 27% in 2050, Figure 1f). This also leads to efficiency improvements driven by electrification that result in an overall reduction of total final energy demand by about 40% relative to 2020 (Ext Data Fig 1), which is less pronounced in AR6 scenarios. Another notable difference is the supply of green hydrogen and e-fuels to replace fossil fuels in hard-to-electrify sectors (Figure 1h). While almost non-existent in AR6 scenarios, their share reaches 9% in the *LeastCost-NZ* and 19% in the *FosFree-NZ* scenario. Generally, ECEMF scenarios are closer to the scenarios of this study, but still show notable differences with respect to penetration of VRE power, electrification and hydrogen particularly for scenarios from models other than REMIND. Through a combination of all the above levers, the *FosFree-NZ* scenario achieves a nearly complete phase-out of fossil fuels by 2050. It reaches this state without relying on large-scale bioenergy production (Figure 1c and i) and achieves EU net-zero GHG emissions by 2050 at considerably lower CCS and CDR deployment in comparison to most AR6 and ECEMF scenarios (Figure 1j and k).

Moreover, we added some sentence about the AR6 comparison to the discussion section:

Our IAM study is the first to explore a complete phase-out of fossil fuels by 2050 in the EU. Scenarios from the AR6 database typically show substantially higher fossil fuel use by mid-century and beyond. First, diverging scenario designs can explain this difference: Unlike AR6 scenarios compatible with 1.5-2°C global warming, our study imposes net-zero GHG emissions in the EU and exogenous limits to CCS. Second, each model has its own characteristics, and REMIND tends to be particularly reactive to climate policies in terms of emissions abatement (Dekker et al., 2023), featuring high renewable power

and electrification shares and limited use of fossil fuels and CCS (Luderer et al., 2022; Soergel et al., 2021). Methodologically, it is important to understand that the full fossil-phase-out is triggered by an endogenous response to increasing carbon prices in the model as a result of limiting the availability of CCS. This contrasts with studies investigating global scenarios with low residual fossil emissions as a result of a broader portfolio of mitigation options (lifestyle changes, advanced technologies) relative to a standard mitigation scenario where these options are not available (Edelenbosch et al., 2024; Fuhrman et al., 2024).

4. There are three ECEMF lines in Figure 1. How does the middle one compare to the authors' Leastcost-NZ and the ECEMF scenario, which is similar to the authors' scenario but allows more fossil fuel? Can the authors compare abatement costs? Here it would help to label the ECEMF lines and dots in Figure 1 for the ones most like Leastcost-NZ and FosFree-NZ.

Thanks a lot for these ideas! We understand your question as to how our scenarios compare to the different ECEMF scenarios. Based on these comments, we adapted Figure 1 to also differentiate models in case of the ECEMF scenarios :

From net-zero to zero-fossil in transforming the EU energy system
Revision 1, Nature Communications

The ECEMF scenarios with residual fossil in the range of our study were also from REMIND, while scenarios with larger residual fossil use were from other models. We adapted the results section in the following way:

Similarly, EU-focused transformation scenarios from the European Climate and Energy Modeling Forum (ECEMF) also indicate residual fossil energy use in 2050 (Figure 1A, blue lines and dots). There

appears to be a model fingerprint as those previous REMIND scenarios tended to be already at a lower levels of residual fossil energy demand than scenarios from other models.

[...]

Generally, ECEMF scenarios are closer to the scenarios of this study, but still show notable differences with respect to penetration of VRE power, electrification and hydrogen particularly for scenarios from models other than REMIND. Through a combination of all the above levers, the *FosFree-NZ* scenario achieves a nearly complete phase-out of fossil fuels by 2050. It reaches this state without relying on large-scale bioenergy production (Figure 1c and i) and achieves EU net-zero GHG emissions by 2050 at considerably lower CCS and CDR deployment in comparison to most AR6 and ECEMF scenarios (Figure 1j and k).

Moreover, we pointed to the abatement cost / carbon prices of the ECEMF scenarios in our discussion section as they tend to be higher than in our scenarios:

In ECEMF scenarios (see Figure 1), EU carbon prices also tend to be around 600 EUR/tCO₂ in 2050, yet in scenarios with higher residual fossil fuel use compared to our *FosFree-NZ* scenario.

We also took up the aspect of different model fingerprints in the discussion section as REMIND has previously contributed scenarios with high renewable penetration and low fossil fuel use and CCS.

Second, each model has its own characteristics, and REMIND tends to be particularly reactive to climate policies in terms of emissions abatement (Dekker et al., 2023), featuring high renewable power and electrification shares and limited use of fossil fuels and CCS (Luderer et al., 2022; Soergel et al., 2021). Methodologically, it is important to understand that the full fossil-phase-out is triggered by an endogenous response to increasing carbon prices in the model as a result of limiting the availability of CCS. This contrasts with studies investigating global scenarios with low residual fossil emissions as a result of a broader portfolio of mitigation options (lifestyle changes, advanced technologies) relative to a standard mitigation scenario where these options are not available (Edelenbosch et al., 2024; Fuhrman et al., 2024).

5. For Figure 2, it is difficult to read some of the font because of the white highlighting and the font not being black. Can the authors use a white background for the font when it is over green or red flows? Even when magnified it is difficult.

We thank the reviewer for the visual suggestions. We worked on the Sankey diagrams to make them easier to read and to compare: the ordering is consistent across scenarios, the font is bigger and more contrasted, and minor links have been removed for clarity.

From net-zero to zero-fossil in transforming the EU energy system
Revision 1, Nature Communications

Figure 2: Energy flows from primary (left) to secondary (middle) to final energy (right) for the EU in 2020 and 2050 in the LeastCost-NZ and FossFree-NZ scenarios. Numbers correspond to the total outflow of energy for primary and secondary energy and total inflow for final energy (buildings, industry, transport) in PWh/yr (thousand of terawatt hours per year). Flows amounting for less than 0.1 PWh/yr have been removed for clarity.

6. Figure 3F should have the same y-axis as Figures 3D and 3E.

Thanks for the suggestion. We chose to leave it as is since the panels refer to different indicators. Only the cross-scenario comparison is relevant in these plots.

7. CCS is mentioned for storage in the introduction and is mentioned for utilization in the results when injection is not used. These two uses should be directly called out in the introduction or combined as CCUS.

CCS stands for carbon capture and storage, while CCU refers to carbon capture and usage in the form of e-fuel production in our scenarios. There may have been a misunderstanding in the paragraph of lines 215 to 222 in the results section. We rephrased it to make this distinction clearer:

Total carbon capture in the EU by 2050 decreases from about 440 MtCO₂/yr in the *LeastCost-NZ* scenario to 260 MtCO₂/yr in *FosFree-NZ*, induced by the limitations on CO₂ storage (Supplementary Figure 2 **Error! Reference source not found.**). Instead of injection into geological storage, captured carbon is increasingly used for e-fuel production. This carbon usage (instead of storage) takes up more than half of the carbon captured in the *FosFree-NZ*.

If this is not the section you referred to, please let us know where this distinction is unclear.

8. What CCS capture rate is used in the model? Capture rates of 95-99% (deep CCS) are currently being funded and actively studied in the literature. While the impact on the Leastcost-NZ case may be low, it will make more bio-genic fuel available and lower overall emissions. A sensitivity run similar to cheap DACSS may be insightful to show that it too does not matter.

We thank the reviewer for this valid comment. The capture rates in REMIND are technology-specific so there is no overall capture rate that we can scale in the model. However, we agree with you that varying the CO₂ capture rate is a meaningful sensitivity in our analysis as captured non-fossil CO₂ has a high value at the net-zero state. Since most of the captured CO₂ in our scenarios comes from biomass gasification (see SI Figure 2) for either hydrogen or liquid hydrocarbon fuel production, we increased its capture rate from the standard value of 90% to 99% in a *HighCR* (high capture rate) sensitivity scenario. As you anticipated, the results only changed slightly and marginal abatement costs in the *FosFree-NZ* scenario decreased a bit (see below). We included this sensitivity together with the sensitivity on DAC costs in a SI Figure as these were the sets of sensitivity scenarios with the smallest difference to the standard scenarios.

From net-zero to zero-fossil in transforming the EU energy system
Revision 1, Nature Communications

Ext Data Fig 1: Marginal abatement cost (Panel A) and aggregate consumption losses (Panel B) of EU net-zero scenarios against fossil fuel carbon consumption for standard scenarios (Standard) and sensitivity scenarios with lower cost of direct air capture (cheapDAC) and higher capture rates of biogenic CO2 capture technologies (HighCR) .

9. In Figure 4, the colors for the legend are washed-out. Please put a black border around the lighter colors so those of us with some color blindness can differentiate them.

Ok, thanks for the notice. We changed it accordingly:

10. In Figure 5, are the authors indicating that DAC cost does not matter, since it overlaps with the standard scenario? This also seems to be true for high imports and high biomass availability. Can the authors comment on this in the paper?

Thanks for your questions on the marginal abatement costs. The reviewer is right that the sensitivity scenarios with low DAC costs only shows small differences relative to the standard scenarios (see Ext. Data Figure 6 below). The cheap DAC scenarios are very similar because DAC is only deployed in the scenario settings with high CCS assumptions (LeastCost-NZ and Intm1-NZ). In the low-fossil scenarios, DAC is neither deployed in the standard nor in the *cheapDAC* setting. However, high biomass availability and high import availability do have a noticeable influence on the marginal abatement cost as can be seen in panel B below (yellow and light green lines). The *FosFree-NZ-HighBiomass* has marginal abatement cost of 570 EUR/tCO₂ relative to 630 EUR/tCO₂ in the standard scenario. At lower FFCC, this gap widens as the *HighBiomass* scenario with maximum CCS reaches down to 280 EUR/tCO₂ compared to 460 EUR/tCO₂ in the standard setting. Similarly, *HighImport* scenarios see lower marginal abatement cost at net-zero (530 EUR/tCO₂ in *FosFree-NZ-HighImport* and 340 EUR/tCO₂ in *LeastCost-NZ-HighImport*) in comparison to the standard scenarios. Hence, the biomass and e-fuel import availability have a stronger impact on the marginal abatement costs than the DAC costs.

Figure 2: Marginal abatement cost and aggregate consumption losses of EU net-zero scenarios with varying levels of residual fossil fuels. A) EU marginal abatement cost on the y-axis against fossil fuel carbon consumption (carbon contained in total demand of fossil fuels) in 2050 on the x-axis across NZ standard scenarios (black line) and corresponding fuel-switching CO₂ prices for replacing fossil liquid fuels by e-fuels based shadow prices from the model (dashed red line). B) EU marginal abatement as in A) across all NZ scenarios (sensitivity scenario in colored lines). C) EU aggregate 2025-2050 consumption losses discounted at 3% per year of all NZ scenarios (sensitivity scenario in colored lines) relative to the Weak Policy scenario (see Methods section M2) on the y-axis against fossil fuel carbon in 2050 on the x-axis. See section M5. Weak Policy Scenario and Consumption Losses **Error! Reference source not found.** for details. See M2. Scenario Assumptions **Error! Reference source not found.** and Ext Data Table 1 for description and categorization of scenarios.

From net-zero to zero-fossil in transforming the EU energy system
Revision 1, Nature Communications

Ext Data Fig 2: Marginal abatement cost (Panel A) and aggregate consumption losses (Panel B) of EU net-zero scenarios against fossil fuel carbon consumption for standard scenarios (Standard) and sensitivity scenarios with lower cost of direct air capture (cheapDAC) and higher capture rates of biogenic CO2 capture technologies (HighCR) .

11. The comparison arguments for Figure 6 are unclear. The nascent nature of these technologies is a formidable issue. It would be helpful to have a reference to the appropriate methods section here to understand how the growth rate comparison is constructed. The note to see Methods in the figure for panel (a) seems to exclude the other panels and these panels would benefit from the direction.

We thank the reviewer for rightly highlighting that one needs to be careful when comparing growth rates of emerging technologies with historical growth rates of technology analogues. In the figure caption of Figure 6, we have now added a reference to the Methods section M6 for each panel. This methods section explains our methodological approach in detail. The depicted growth rates are historical compound annual growth rates (CAGRs). Kindly note that we only use these historical growth rates after 2030 to make sure that the nascent technologies have been deployed enough such that it is warrantable to use yearly growth rates at all

12. In looking at the figure the first question that comes to mind is “Are the solar and wind growth rates based upon projected capacity and scaled to plot as done in Panel d, or are they based on projections that are derated for capacity factors?”. The idea is right, but the energy comparison seems wrong. A useful comparison to put it in context may be to oil refineries. How many e-fuel facilities of a given capacity are required each year to reach the energy demanded and how many refineries are built each year or what percentage of the total European or world refineries does this capacity represent? What capital expenditure does this represent? Also, are the authors indicating that it will be difficult for the Leastcost-NZ scenario to meet the required output because the projected energy growth rate is below that for wind? If so, I missed the important point.

The reviewer is right to ask these questions. All growth rates shown (in panel b and d) are compound annual growth rates (CAGRs) of the corresponding historical production (in TWh/year) for the past 20 years (17 years for tight oil due to data availability). They are not projections, but empirically realised growth rates in the past. If we assume that capacity factors have not changed substantially for these technologies, the CAGR of the installed capacity (in GW) would yield the same value.

We like the reviewer's idea to compare the growth of e-fuels to the historical growth of oil refineries. In the revised manuscript, we have now replaced the previous comparison to hydro power with a comparison of the 20-year CAGR of global oil refinery capacity (from 1965-1985, the period in which it grew at its fastest in the EI Statistical Review of World Energy 2024). While the CAGR for hydro was 2.4%/year, the CAGR of oil refining capacity is 3.8%/year. Although we would like to note that oil refineries are not directly comparable with the e-fuel production chain because of additional production steps in the latter case (electricity, hydrogen, carbon capture), we agree with the reviewer that this is a useful point of comparison.

Regarding the comparison to the LeastCost-NZ scenario, the reviewer is right and did not miss the important point. The main point of this figure is to compare the required upscaling of e-fuels and CCS in the LeastCost-NZ and the FosFree-NZ scenario with each other. In the LeastCost-NZ scenario, the required e-fuel production could be satisfied if e-fuels grow as fast as wind power historically has. In the FosFree-NZ scenario, e-fuels would instead need to grow approximately as fast as solar power historically has.

13. While e-fuels can be imported as needed, the paper concerns primarily EU e-fuel, so a stronger argument can be made if Figure 6(a,b) only represent EU demand and supply.

We thank the reviewer for bringing up the regional scope of the scale-up analysis. However, particularly for liquid e-fuels, which are easy and cheap to transport using existing infrastructure, the EU will likely import a substantial share from regions with more favorable renewable potentials in the future, instead of relying on domestic production. Therefore, we would argue that the comparison between global e-fuel supply and EU e-fuel demand is meaningful for our scenarios. That global supply exceeds EU demand is a necessary condition for these scenarios to be feasible. As we write in the text, feasibility furthermore requires that the EU can secure large shares of the global e-fuel market. Alternatively, we could have assumed that a certain share of the global market goes to the EU. However, this seemed like a less transparent option to make this comparison as this share is quite uncertain. Based on your remarks, in the revised manuscript we have now added additional annotations in panel b in order to more clearly convey the comparison between potential global e-fuel supply (based on historical growth rates of other technologies) and the required EU e-fuel demand from our scenarios (see below).

From net-zero to zero-fossil in transforming the EU energy system
Revision 1, Nature Communications

Figure 3: Scenario-dependent trade-off between scaling-up global e-fuel production or European CCS capacity. **a**, Global e-fuel project announcements by status. By 2030, we assume that 10 TWh/yr of global e-fuel production capacity can be realized, corresponding to the aviation and maritime e-fuel sub-quotas legislated by the EU (see Methods M6). **b**, Scaling-up of global e-fuels until 2050 following historical global 20-year growth rates of solar (39 %/a) and wind power (20 %/a) from 2003-2023, and oil refining capacity from 1965-1985 (5 %/a, see Methods M6). Note that the depicted comparison of energy production shows the historical 20-year growth rate of the corresponding technologies starting from 10 TWh/yr in 2030 and not the exact historical time series of these technologies. Global e-fuel production would need to grow as fast as solar PV and a large share would need to be sold to the EU in order to meet EU e-fuel demand in the FosFree-NZ scenario. **c**, European CCS project announcements by status. By 2030, we assume that 25 MtCO₂/yr of European CCS capacity can be realized, corresponding to the 88% failure rate in Kazlou et al. (2024) for the year 2030 (see Methods M6) **d**, Scaling-up of European CCS until 2050 following historical global 20-year growth rates of global solar PV (39 %/a), total US oil production from 2003-2023 (5 %/a), as well as the 17-year growth rate of US tight oil production from 2007-2024 (20 %/a, see Methods M6). European CCS capacity would need to grow almost as fast as US tight oil in order to meet the requirements in the LeastCost-NZ scenario, whereas the FosFree-NZ scenario keeps CCS at a minimum.

14. These arguments are stronger for CCS for solar and wind and how building refineries may be more appropriate. I don't think oil production in the US is a good comparison. All of these technologies have issues with growth rates increasing because of subsidies, changes in government policy to increase scale, and demand-pull versus technology-push. If the authors want to bound the required growth rates, taking the historical technology growth rate and plotting it is fine, but label it as a 20% or 5% growth rate rather than the specific technology.

We agree with the reviewer that the comparability of the shown technologies may be limited due to different policy support, technology characteristics (e.g. modularity) and competitiveness. We

deliberately do not make a point about which technology is a particularly suitable analogue for CCS. The primary goal of the technology comparison is an illustration of the required scales of ramping-up CCS. In this context, US tight oil production serves as a proxy for a technology comparable to CCS in terms of infrastructure and limited modularity (requires exploration, drilling, large-scale site-specific construction). Tight oil production in the US was expanded quite rapidly in the past two decades which shows that growth rates of this level are feasible for these types of projects. The reviewer is right that these technologies primarily serve as an illustration of different growth rates.

In the figure caption of the revised manuscript, we now state the CAGR of all depicted technologies explicitly. We thank the reviewer for highlighting this point. However, we think that it is useful to keep the specific technologies as a reference point for readers, instead of just mentioning the CAGR.

15. As a comparison, Hanna, Abdulla, Xu, and Victor have a paper in Nature Communications that describes the emergency deployment of DACS and compares the required build rate and money with historical examples. Solar is used in this paper but as a reference high growth rate that was also achieved with US Liberty ships in WWII. Hanna, R., Abdulla, A., Xu, Y., & Victor, D. G. (2021). Emergency deployment of direct air capture as a response to the climate crisis. Nature communications, 12(1), 368.

We thank the reviewer for pointing us to this paper. It makes sense to think about the massive e-fuel growth rates required in the full fossil phase-out scenario as a case of emergency deployment. We included the following sentence in the section on feasibility:

Given the complexity of the e-fuel supply chain including green hydrogen production and non-fossil carbon capture, these growth rates imply a crisis-like emergency deployment as investigated by previous feasibility studies on electrolysis or direct air capture (Hanna et al., 2021; Odenweller et al., 2022).

16. Is the 88% failure rate extrapolated to 2050 given learning rates are involved in the process? If so, this is unlikely. Yes, for nascent technology the first efforts may fail, and we have seen CCS facilities shutdown because the price support is no longer there (WA Parish) or the cost overruns are excessive (Kemper). However, these failure rates will decrease as the learning rate improves and causes of the failure are removed (policy, financial, and technical). If the failure rate is extrapolated, a discussion of the Pareto of failures would be helpful. Rubin, Herzog, Mac Dowell, Heuberger are good sources for learning rates.

We would like to apologise for the confusion caused. We only use the empirical 88% failure rate from Kazlou et al. to derive a plausible value of CCS deployment for 2030. We do not use it beyond 2030. We fully agree that the failure rate is likely to decrease in the future as the technology matures. In the revised manuscript, we have edited the figure caption to remove any ambiguities about the use of the failure rate.

17. The line colors in Figure 6 Panel d are hard to discern. Can the authors use bolder colors for the thin lines?

We agree. In the revised manuscript we have improved the coloring of the lines.

18. In line 329 the authors mention the pressure on CCS is relieved with more e-fuel. Do the authors ever recommend multiple approaches to achieve a higher likelihood of achieving net zero? This is done in the last paragraph and should be linked back to this line and thought.

Thanks for this comment. We overall revised the discussion section. We rephrased the formulation around CCS and CCU development to avoid any ambiguities:

In addition, substantial carbon capture of more than 200 MtCO₂/yr is required in all net-zero scenarios even as CO₂ storage is limited in *FosFree-NZ*. Investing in non-fossil carbon capture is therefore essential as it is needed for both e-fuel production and CDR via CO₂ storage. E-fuel production and CO₂ storage technologies should be developed in parallel to create robust pathways that secure against potential delays in up-scaling one of them.

19. Total energy in Ext Data Fig 1 does not show non-heating energy.

Thanks for noticing. The distinction is actually not critical in this plot and it is more complicated to differentiate our industry electricity demand into heating and non-heating purposes. We therefore decided to remove this distinction overall and only show total electricity demand.

20. SI 4 should have the same y-axes.

Thanks for the idea. However, choosing the same y-axes for all panels made the scenario differences very hard to distinguish. We therefore left the variable axes as they are.

Grammatical editing to improve the paper

Thanks for the close look. We included most of the points. We left a comment in case we did not follow your suggestions.

1. Line 9 should read “Quantifies the additional effort...” instead of “Quantification of the additional effort...”

2. Line 11 should read “By 2050, least-cost net-zero scenario abates 87% of EU fossil fuel use with increased renewable power, electrification, and biofuels”. Last comma is missing after electrification in original if the authors don’t use suggest change.

3. For the third bullet, the detail about the oil-based hydrocarbons is not required for the highlight. This bullet may be shortened to “Carbon-neutral e-fuels are crucial to replace the remaining 13% of hardest-to-abate fossil fuels used for international transport and chemicals”.

Thanks, we left it as is. For the editor to see if bullets are needed.

4. For line 16, the “(500-1000 €/tCO₂)” can be removed because this is a highlight, and the detail can be noted in the main text. Similarly, “roughly” is not required. The authors can also remove “a” prior to least-cost and replace it with “the” because the second bullet introduces this scenario.

Thanks, we did not change it for now. Depends on whether highlights will be included in the final version.

5. Line 24 should have a comma after aviation.

6. In line 25, “about” can be omitted.

7. In line 26 the expect range can also be omitted, as can “by 2050”. The range is presented in the body and 2050 is already mentioned twice in the abstract.

8. In line 27, omitting “also” increases the tension and make the abstract more dramatic.

9. In line 34, replace “About” with “Almost” and hyphenate three-quarters.

10. While EU is a well-known acronym, I don’t recall the journals policy for requiring to define acronyms like this or CO₂. Greenhouse gas emissions is defined. Same with COP28, etc.

11. In line 41 the authors introduce CCS and later on the authors mention captured carbon being used for e-fuels. Is there ever a distinction between CCS and CCUS (carbon capture for utilization and storage)?

We do not use the umbrella term CCUS to always distinguish clearly between carbon capture and storage (CCS) and carbon capture and usage (CCU). Carbon usage for e-fuels is CCU.

12. In line 46, replace “fast” with “rapid”. Also, if the authors are including DACS in this thought, an appropriate reference for this is Hanna, Abdulla, Xu, and Victor, “Emergency deployment of direct air capture as a response to the climate crisis.”

Thanks. We refer to geological CO₂ storage scale-up and scenarios from Integrated Assessment Models.

13. In line 46, remove the comma after “aspects” and replace it with “by”. To emphasize the overestimation, the comma in line 47 can be removed and “which” replaced with “that”. The overestimation is an important part that should be emphasized.

14. In line 50, “false” is a better word than “wrong”.

15. In line 50, it should be “energy-sector stakeholders scale-down mitigation ambitions in anticipation of sufficient CCS or CDR deployment to abate emissions.”

16. Line 57 should “is to” instead of “would be to”.

17. Line 58 should be “in order to” instead of “that would”.

18. Line 60 needs a comma after “decisions”.

19. Line 69 omit “also”.

20. In line 70 replace “they” with “these studies”.

21. Line 79 should read “, and derive abatement cost curves for the remaining 10-15% of fossil emissions...”.

22. In line 80 the authors can omit “EU” because the authors just said so but why do the authors change from sector to system? Sector is preferred for consistency.

23. In line 94, change to “lowest level at which net-zero can be achieved with the model...”. My assumption from the sentence is that this is a limitation of the model.

We followed your suggestion. However, while there certainly are limitations of the model in investigating residual emissions abatement (see discussion section), we would like to stress that this is already quite a low level of CCS deployment (also relative other scenarios from AR6 or ECEMF). Going even lower with CCS deployment at net-zero would require higher deployment of other CDR options to offset residual emissions in agriculture and industrial processes.

24. In line 97, break this up into two sentences. “We also examine sensitivity scenarios with high...”.

25. In line 98, add a comma after “(highBio)”.

26. In line 203 replace “they” with “our EU NZ scenarios” or something like this to differentiate the authors work from the IPCC work.

27. In line 108 add “the” after “of”.

We left it as is.

28. In line 114 add a comma after transport.

29. In line 115 start a new sentence with “This is done by increasing the share of renewable energy in the electricity energy mix to 60% by 2050...”.

We left it as is.

30. In line 117, is 27-30% in 2050? The discussion on efficiency improvements should be a new sentence because it is long and complicated.

31. In line 119, “substitute” should be “replace”.

32. Line 144 should say “Fossil fuels are largely replaced with renewable-based electricity.” Directly and indirectly doesn’t add anything and biomass isn’t typically considered as renewable energy.

We left it as is. We would like to highlight direct and indirect electrification as two different ways to use electricity for decarbonization in this context. Moreover, we respectfully disagree to the statement that biomass cannot be regarded as renewable energy.

33. Line 157 should be “, and de-fossilization of...”.

34. In line 160, do the authors mean higher renewable penetration? If so, this phrase is better. (“...to one with high renewable penetration that reduces coal consumption.” If one needs to, one can also add “high renewable penetration of variable solar and wind capacity that...”)

We rephrased the section to

First, at relatively low marginal cost of abatement, the power sector already transitions until the mid-2030s to a system based on VRE from wind and solar power (Ext Data Figure 2). This allows for a phase-out of coal power and in turn a strong reduction of coal consumption.

We wish to keep variable renewable energy and its definition “energy from wind and solar power” in the sentence to be able to use the abbreviation VRE in the following parts.

35. In line 162 is natural gas and oil replacing coal, or is something replacing all three?

We rephrased:

Second, direct electrification of final energy across end-use sectors unfolds up to 2040 (Figure 3e) as electricity replaces mainly natural gas and oil for energy use in low-temperature heating and road transport. The consumption of natural gas and oil decreases less rapidly than for coal, though (Figure 3b and c).

36. In line 164 the “most expensive step” seems lost. If the authors wish to emphasize the time-step and the expense-step, it can be added after “finally”. It seems unnecessary to have this clause though.

37. In line 189, can be simplified to “which is a reduction of 99.5% from the 2020 level.”

38. Line 196 should have a comma inserted “liquid fuels), and...”

39. Line 198 should have a comma inserted “district heating, and domestic transport sector...”

40. Line 199 should be broken into two sentences. “even in LeastCost-NZ. Furthermore, fossil emissions in the buildings sector are largely reduced...”

41. In line 202, “substitute” should be changed for “replace”.

42. In line 203 adding parentheses makes it easier to read. “Primarily, e-fuels (i.e. synthetic liquid fuels or chemicals produced from electrolytic hydrogen (Ext Data Fig 5) and non-fossil CO₂ (Ext Data Fig 6))...”

43. Line 205 needs a comma after “aviation”.

44. Line 208 is confusing. Do the authors mean “Bioenergy is used to produce hydrogen in the LeastCost-NZ scenario, and is used in the FosFree-NZ scenario to produce biogas that replaces fossil gas in industry and buildings (SI Figure 1).”

45. Line 219 should be rewritten. “Conventional CCS fails to reach the 2050 LeastCost-NZ scenario-specific limit of 2000 MtCO₂/yr, as less than 400 MtCO₂/yr is captured. However, the lower scenario-specific limit of 110 MtCO₂/yr is reached in the FosFree-NZ scenario.

46. “Fossil-fuel Carbon Consumption” should be hyphenated in line 227.

47. In line 245 marginal abatement costs can be replaced with MACs or “...MAC is quite...”
48. Line 246 should be “higher biomass availability...” and decrease should be decreases.
49. Since the authors defined MAC as singular, all uses of it should be singular—the verb needs an s.

Thanks. We define it as plural now.

50. Line 249 should read “...indicates that the CO₂ prices...”
 51. Line 279 should be made into two sentences. The second sentence can begin “Of this required quantity, less than 1%...”
 52. Line 281 should have a comma after 2030.
 53. Line 282 is unclear and needs to be rewritten as two sentences.
 54. There should be a comma after electrification in line 316.
 55. In line 315, “in time” is not required after “consecutively” because it is implied.
 56. Line 317 is too long and detailed. Please rewrite.
 57. In line 317, “already quite impactful” is too informal. Perhaps “...are significant for achieving net-zero emissions: Decarbonization and electrification result in decreasing 2050 EU fossil primary energy consumption by 87% relative to 2020 levels at a marginal abatement cost of 300 EUR/tCO₂. This reduction is driven by a transition...”
 58. When the authors add “...and some bio-based fuels” there needs to be a comma before and. Also, isn’t this the third step a contradiction to the statement that this covers only the first two steps? The rest of the sentence is more confusing.
 59. Hyphenate fossil-fuel carbon consumption in line 319.
 60. In line 320-321, at what point? “...moving from a least-cost to a fossil-free net-zero scenario requires that the focus changes from (cost solutions of some sort?) to supply-side energy solutions such as non-fossil hydrocarbons.”
 61. In line 321, the secondary clause should be move to the beginning of the sentence.
 62. In line 323 add “the” before “remaining”.
 63. Hyphenate one-third in line 334. The journal style guide may differ.
 64. In line 339, replace “but” with “and”.
 65. In line 340, specify green hydrogen or make it clear in the results that the authors are only considering green hydrogen.
 66. Line 346 should have a comma after “economy”. As much as what?
 67. In line 350 the authors can emphasize fossil-free target to link to their scenario.
- We left it as is.
68. In line 351 there should be a comma after “fossil use”.

69. In line 353, the authors can emphasize the obstacles for CCS by removing the “which” and replacing it with “that”. Otherwise, they need to add a comma.

70. In line 357, the sentence will have greater emphasis if the authors change it to “...TWh/yr—ten times the current EU liquid fuel consumption in aviation and shipping.”

It is about the current fossil liquid fuel consumption. Made this more precise:

This is mainly driven by the need to rapidly scale-up e-fuel supply to 1000 TWh/yr in 2050, which is about as much as the total current EU fossil fuel consumption in aviation and shipping.

71. In line 357, “They” should be changed to “These e-fuels”.

72. Line 359 should read “...that only allow a certain percentage of sector-specific fossil fuel use...”.

#	Lines	Section	Comment
1	20	Abstract	You have some space to specify “ REMIND integrated assessment model” which is useful for IAM-readers.
2	26-27	Abstract	Reinforce that it is a complete fossil phase-out so it is not mistaken by general readers as the EUs current plans. Also, consider the following the sentence structure that might be more impactful. “Although posing additional transformation challenges, committing to a complete fossil phase-out would strengthen EU climate policy”
3		Introduction	Double-check the referencing format of EU-related citations. There is an interchange between the EU, European Council, and EU Commission. These may be correct, but these documents can often be incorrectly referenced; the EU might need to be spelled out to “European Union”.
4	38	Introduction	Perhaps it is more clear to write “However, this ambition has not translated into concrete EU targets for completely reducing-or phasing-out fossil fuel consumption.” Since the EU Green Deal and related material do suggest fossil phase-out by adoption of several renewable technologies etc. it is perhaps in the paper's interest to focus on the complete phase-out aspect.
5	44	Introduction	Check if there is an accidental additional space between “available remains”
6	51	Introduction	This is a great sentence, but it is also a generalisation. Are you able to pull a citation or case-study example to strengthen this? If not, add some caution to the sentence such as: “This poses the risk of creating wrong expectations as stakeholders in the energy sector may scale down mitigation ambition in anticipation of sufficient future CCS or CDR to abate their fossil emissions.”
7	87-99, 647	Results, Methods	Please can you justify why, in the LeastCost-NZ scenario, the EU land carbon sink is assumed to jump to 370 MtCO ₂ /yr in 2050 suddenly, but in the FosFree-NZ scenario, the land carbon sink remains at 240 MtCO ₂ /yr in 2050? This seems like an assumption that should be consistent; it implies that the LeastCost-NZ scenario will have substantial changes in biomass, soil carbon, forests, etc., immediately in 2050. Moreover, both scenarios have the same biomass consumptions in Figure 11

			A substantial 130 MtCO₂/yr (+50%) sudden increase likely impacts the solution space for REMIND. Please provide substantial justification for this assumption with evidence or sensitivity testing of the value at 240 MtCO₂/yr etc. “First, we model a least-cost NZ scenario (LeastCost-NZ) where CO₂ storage injection is free to scale up to 2 Gt/yr in the EU at low injection costs. Furthermore, we optimistically assume that the EU land carbon sink can increase from the current level of 240 MtCO₂/yr to 370 MtCO₂/yr in 2050, exceeding the 2030 goal of 310 MtCO₂/yr (European Commission, 2021). This is our benchmark net-zero scenario without a fossil phase-out. Second, we model a fossil-free NZ scenario (FosFree-NZ) where CO₂ storage injection is limited to 110 MtCO₂/yr, which is the lowest setting for the model to still reach the net-zero target, with the land carbon sink remaining at 240 MtCO₂/yr.” “We make an optimistic assumption of a carbon removal of 370 MtCO₂/yr by 2050 within the range of Pilli et al. (2022), which is slightly larger than what was used for the EU Commission (2018).”
8	101-110	Results, Figure 1	As discussed in the text, the FosFree-NZ and LeastCost-NZ scenarios have considerably more optimistic fossil reductions compared to the 1.5-2.0C pathways. The key drivers of this are discussed in the following paragraph and Figure 1. However, in the discussion section, please discuss the model factors and assumptions contributing to these differences compared to other models that feed the IPCC scenarios. This can be at a general level, but the study results are based on the assumptions and limitations of REMIND, and it's an important discussion point. One example is that REMIND is based on perfect foresight and long-term decisions, meaning it is more likely to select the most optimum and optimistic technology solutions compared to other models (e.g. rapid solar, electrification). In contrast, IMAGE has myopic foresight is focused on short-term decisions and is more likely to lead to more pessimistic solutions. Below, are a few references that could help with this discussion (Please note, I have no relationship to these articles) Dekker, M.M., Daioglou, V., Pietzcker, R. et al. Identifying energy model fingerprints in mitigation scenarios. Nat Energy 8, 1395–1404 (2023). https://doi.org/10.1038/s41560-023-01399-1 Mathijs Harmsen et al 2021 Environ. Res. Lett. 16 054046 Integrated assessment model diagnostics: key indicators and model evolution 10.1088/1748-9326/abf964

			I Keppo et al 2021 Environ. Res. Lett. 16 053006 Exploring the possibility space: taking stock of the diverse capabilities and gaps in integrated assessment models 10.1088/1748-9326/abe5d8
9		Figure 2	This is a nice Sankey, but it is slightly hard to follow and compare between the three. I would suggest to ensure the ordering on the left and the right stays the same to help readers compare. For example, ensure the fossil source (leftmost) stays the top, and apply to the others. Moreover, in the EU27 LeastCost-NZ 2050 (Middle), the final part of the secondary energy value for hydrocarbons is hidden (248_), please ensure this is fully visible.
10		Figure 3	This is an insightful figure that clearly illustrates the scenario differences, particularly from 2035 which seems to be a critical point. It could benefit from an additional chart related to CCS/CDR deployment and could perhaps be included on a secondary axis. It would help understand CCS/CDR deployment over time.
11	205-206	Results	Please could you check and clarify that this sentence makes sense. I'm not sure what you mean by e-fuels being deployed once CDR becomes limited? “As e-fuels are an expensive mitigation option, they are only deployed at scale once CDR options like BECCS and DACCS are limited, which aligns with previous research”
12	235-270	Results, Figure 5	Focussing on the leftmost chart, there is a substantial MAC jump from ~250 to ~500 EUR/tCO₂, for limiting FFCC from 400 to 300 MtCO₂/yr. What is the reason for the substantial initial jump in MAC in the standard model? In general, I think this section could be strengthened with a few sentences to describe the MAC trends (initial spike, plateau, spike again – from higher to lower FFCC) rather than just from the end-to-end.
13		Discussion	The discussion could be significantly strengthened, as it mainly summarises the results. The study results are important, can have a significant impact, and could land much more effectively with an improved discussion. Here are some suggestions to consider:  • A quick summary is fine, but I think it could be shorted. • 338-347 discusses some assumptions and limitations nicely, but I would recommend including discussion on the uncertainty and limitations of using only REMIND (see comment 7+8), and how outcomes can

			differ from different IAMs. Moreover, it may be useful to discuss the assumptions on land carbon sink, and what other parameters could have been sensitivity testing, and uncertainty of regional disaggregation since you mention REMIND contains 8 EU regions in the methodology.  • Can you discuss and draw some key comparisons compared to other literature mentioned in the introduction? A multi-model analysis of the EU's path to net zero, Boitier, Baptiste et al. Joule, Volume 7, Issue 12, 2760 – 2782 https://doi.org/10.1016/j.joule.2023.11.002. I was also surprised, this paper was not included, which can provide some relevant discussion points too. Ueckerdt, F., Bauer, C., Dirnaichner, A. et al. Potential and risks of hydrogen-based e-fuels in climate change mitigation. Nat. Clim. Chang. 11, 384–393 (2021). https://doi.org/10.1038/s41558-021-01032-7. • The final paragraph discusses implications but also shies away from making clearer and meaningful recommendations that policy-makers or practitioners can take away. The core of the paper is attempting to show pathways to zero fossil, and it would be beneficial at the end to have a few bullets or numbered items on key recommendations/suggestions e.g. strengthening support/investment/import for carbon neutral fuels development for hard-to-decarbonise sectors.
14	552	Methods	I think this is the more relevant definition “ REMIND is an energy-economy-environment model ” since REMIND and other IAMs contain environment modules etc.
15	729	Methods	I think you mean 2030? “ This scenario reduces total GHG emissions to only about 2200 MtCO₂/yr by 2050 2030, which is about the level of 55% reductions relative to 1990 envisaged for 2030 with the regulation of the European Green Deal. ”